# Genomic mosaicism with increased amyloid precursor protein (*APP*) gene copy number in single neurons from sporadic Alzheimer's disease brains

Diane M Bushman[1,2†], Gwendolyn E Kaeser[1,2†], Benjamin Siddoway[1], Jurgen W Westra[1‡], Richard R Rivera[1], Stevens K Rehen[1§], Yun C Yung[1], Jerold Chun[1*]

[1]Department of Molecular and Cellular Neuroscience, Dorris Neuroscience Center, The Scripps Research Institute, La Jolla, United States; [2]Biomedical Sciences Graduate Program, University of California, San Diego, La Jolla, United States

**\*For correspondence:** jchun@scripps.edu

[†]These authors contributed equally to this work

**Present address:** [‡]Genedata Inc., Lexington, United States; [§]National Laboratory for Embryonic Stem Cell Research, Federal University of Rio de Janeiro, Institute of Biomedical Sciences, Rio de Janeiro, Brazil

**Competing interests:** The authors declare that no competing interests exist.

**Abstract** Previous reports have shown that individual neurons of the brain can display somatic genomic mosaicism of unknown function. In this study, we report altered genomic mosaicism in single, sporadic Alzheimer's disease (AD) neurons characterized by increases in DNA content and amyloid precursor protein (*APP*) gene copy number. AD cortical nuclei displayed large variability with average DNA content increases of ~8% over non-diseased controls that were unrelated to trisomy 21. Two independent single-cell copy number analyses identified amplifications at the *APP* locus. The use of single-cell qPCR identified up to 12 copies of *APP* in sampled neurons. Peptide nucleic acid (PNA) probes targeting *APP*, combined with super-resolution microscopy detected primarily single fluorescent signals of variable intensity that paralleled single-cell qPCR analyses. These data identify somatic genomic changes in single neurons, affecting known and unknown loci, which are increased in sporadic AD, and further indicate functionality for genomic mosaicism in the CNS.

## Introduction

The genome has been classically viewed as being constant from cell to cell in the same individual, with genomic differences passed on through the germline. However, within neurons of the brain, numerous studies have reported somatic variability producing complex genomic mosaicism but having unknown function. Identified forms of somatically arising genomic mosaicism include aneuploidy (reviewed in *Bushman and Chun, 2013*), LINE elements (*Muotri et al., 2005*; *Baillie et al., 2011*; *Evrony et al., 2012*), copy number variations (CNVs) (*Gole et al., 2013*; *McConnell et al., 2013*; *Cai et al., 2014*), and DNA content variation (DCV) (*Westra et al., 2010*; *Fischer et al., 2012*).

AD is the most common form of dementia and is characterized by the presence of amyloid plaques, synaptic loss, and cell death (*Alzheimer's Association, 2013*), notably affecting the prefrontal cortex. The major component of these plaques is β-amyloid (Aβ), a protein encoded by *APP* (*Goldgaber et al., 1987*; *St George-Hyslop et al., 1987*; *Tanzi et al., 1987*). Familial AD accounts for less than 5% of all cases and has been genetically linked to mutations in *APP* and two presenilins (PSEN), PSEN1 and PSEN2, the catalytic components of γ-secretase and the units responsible for cleavage of APP (*Price and Sisodia, 1998*; *Bertram et al., 2010*). In addition, *APP* gene dosage is strongly associated with AD pathogenesis based on multiple lines of evidence. First, Down syndrome (DS), with three copies of *APP*, produces neuropathology virtually identical to AD (*Glenner and Wong, 1984*; *Delabar et al., 1987*) and *APP* locus duplications are sufficient to cause familial AD (*Rovelet-Lecrux et al., 2006*; *Sleegers et al., 2006*; *McNaughton et al., 2012*). Moreover, AD-protective effects have been reported in DS with *APP* deletion via partial trisomy (*Prasher et al., 1998*), as well as in familial AD

**eLife digest** The instructions for living cells are contained in certain stretches of DNA, called genes, and these instructions have been largely considered to be invariant, such that every cell in the body has the same DNA. However, research has revealed that many neurons in the human brain can contain different amounts of DNA compared to other cells. When cells with varied DNA are present in the same person, it is referred to as mosaicism. The effects of this mosaicism are unknown, although by altering the instructions in brain cells, it is suspected to influence both the normal and diseased brain.

The brains of patients with Alzheimer's disease often contain deposits of proteins called amyloids. The precursor of the protein that makes up most of these deposits is produced from a gene called the amyloid precursor protein gene, or *APP*. Having an extra copy of the *APP* gene can cause rare 'familial' Alzheimer's disease, wherein the *APP* duplication can be passed on genetically and is present in all the cells of a patient's body. By contrast, 'sporadic' Alzheimer's disease, which constitutes around 95% of cases, does not show any difference in the number of *APP* genes found in tissue samples, including whole brain. The early studies that discovered this were conducted before an appreciation of brain mosaicism, and thus single neurons were not investigated. This raises the possibility that the number of *APP* genes may be mosaically increased, which would not be detected by examining non-brain or bulk brain tissue.

Bushman, Kaeser et al. used five different types of experiments to examine the DNA content of single neurons and investigate whether mosaicism could explain the discrepancy in the results of the previous studies. The neurons from people with Alzheimer's disease contained more DNA—on average, hundreds of millions of DNA base pairs more—and more copies of the *APP* gene, with some neurons containing up to 12 copies.

Bushman, Kaeser et al.'s findings present evidence of a way that mosaicism can affect how the brain works by altering the number of gene copies, and how this impacts the most common form of Alzheimer's disease. Many questions arise from the work, including when does mosaicism arise, and what promotes its formation? How does this relate to age? What parts of the genome are changed, what genes are affected, and how do these changes alter neuronal function?

Furthermore, Bushman, Kaeser et al.'s work suggests that mosaicism may also play a role in other brain diseases, and could also provide new insights into the normal, complex functions of the brain. In the future, this knowledge could help to identify new treatments for brain diseases; for example, by identifying new molecular targets for therapy hidden in the extra DNA or by understanding how to alter mosaicism.

with an *APP* partial loss-of-function mutation (*Jonsson et al., 2012*). However, seminal studies in the 1980s failed to detect evidence of *APP* amplification in sporadic Alzheimer's disease peripheral blood and whole brain (*Podlisny et al., 1987*; *St George-Hyslop et al., 1987*; *Tanzi et al., 1987*) despite strong linkage in familial AD, thus linkage between sporadic AD and *APP* remains unclear.

The existence of region specific genomic mosaicism in the normal brain (*Westra et al., 2010*) raised the possibility that DCV, defined as variations in the total DNA amount present in a single cell or population, might play a functional role in sporadic brain diseases by altering pathogenic loci in individual cells. The validated pathogenicity of *APP* in familial AD suggested that mosaic alterations in *APP* copy number within single neurons may play a role in producing sporadic AD. Through the use of five independent experimental approaches, we report increased somatic genomic variation within individual sporadic AD neurons involving mosaic increases in both DNA content and *APP* copy number.

## Results

Methodologies to identify genomic mosaicism in AD brain nuclei utilized multiple independent approaches (*Figure 1*). First, neuronal nuclei were isolated from paired prefrontal cortex and cerebellum of postmortem human brains as previously described (*Figure 1A*) (*Westra et al., 2008*, *2010*). Nuclei were analyzed for: DNA content changes (*Figure 1B*), *APP* CNVs using qPCR in small populations (*Figure 1C*) and single-nuclei (*Figure 1D*), trisomy 21 using standard fluorescence in situ hybridization (FISH) (*Figure 1E*), and amplified *APP* loci using PNA-FISH (*Figure 1F*).

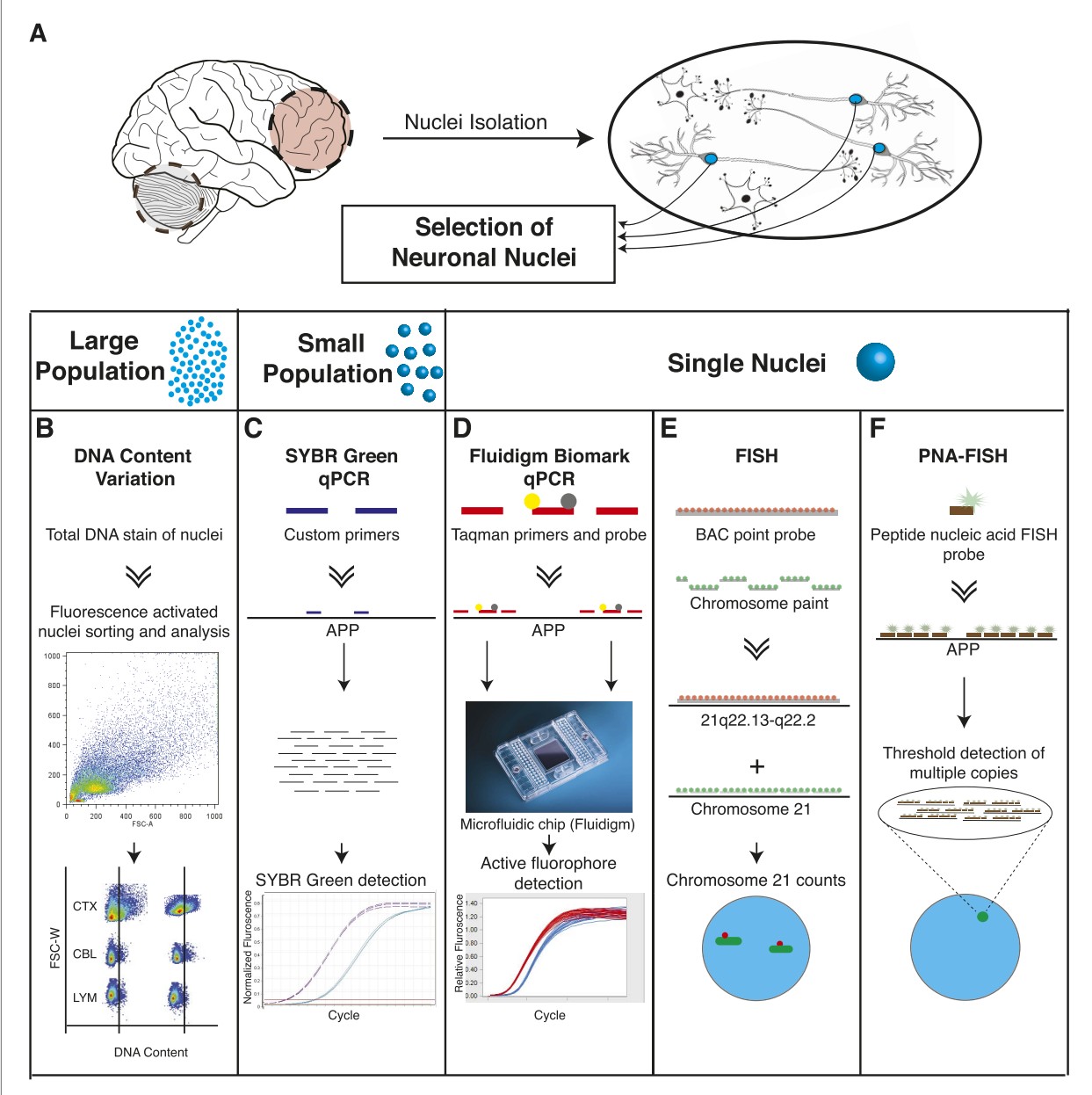

**Figure 1**. Methodologies used in assessing genomically mosaic AD. (**A**) Neuronal nuclei were isolated from the prefrontal cortex and cerebellum of postmortem human brain (see 'Materials and methods' for samples used) as described (*Westra et al., 2008*). (**B**) Nuclear DNA was stained with propidium iodide (PI) and DNA content was quantified using flow cytometric analysis. (**C**) *APP* copy number variations were analyzed in small populations of nuclei (~75 genomes) using custom primers for exon 14 of *APP*. (**D**) Single-cell qPCR assessed *APP* copy number variations in individual neuronal nuclei via TaqMan probes and a modified Biomark integrated fluidic chip system (Fluidigm Corporation, South San Francisco). (**E**) FISH paints against the whole q arm of chromosome 21 and a point probe against a region on the q arm of 21 (21q22.13-q22.2) were used to double-label and call aneusomies in AD samples. (**F**) Peptide nucleic acid (PNA) FISH was combined with super resolution microscopy for threshold detection of *APP* copy number above ~2 occurring at a single locus.

## AD cortices show increased DNA content variation (DCV)

Previous work identified DNA content changes with regional variability in the non-diseased brain, where prefrontal cortical nuclei—particularly neuronal nuclei—displayed increased DCV compared to nuclei of the cerebellum and non-brain controls (*Westra et al., 2010*). The current study utilizes the same techniques for DNA content analysis by flow cytometry using propidium iodide (PI). PI staining

is the predominant methodology for quantitatively differentiating nuclei or cells with variable DNA content and is routinely utilized as a gold standard in multiple fields including genomic comparisons across species in botany (*Dolezel et al., 2007*), studies of the cell cycle (*Krishan, 1975*), and DNA degradation produced by apoptosis (*Riccardi and Nicoletti, 2006*). Prior analyses ruled out the effects of DNA dyes, nuclear size, mitochondrial contamination, and autofluorescent lipofuscin on DNA content and validated genomic increases in DNA content using quantitation of CENP-B PNA probes against human centromere repeats (*Westra et al., 2010*) which have been further substantiated by identification of copy number gains in single human neurons (*Gole et al., 2013*; *McConnell et al., 2013*).

To further validate DCV in human neurons, whole genome amplification (WGA) was used on cortical neuronal nuclei sorted into populations of high or low DNA content based upon PI intensity. Nuclei with high or low DNA content were sorted into 12 replicates of 1000, 500, or 100 and were then subject to DNA content assessment by WGA to assess starting amounts of DNA template in each sample (*Figure 2A*). Nuclei were denatured and amplified by multiple displacement amplification (MDA) during which DNA synthesis was continually measured by SYBR Green fluorescence (*Figure 2B*). In every case, nuclei with high PI intensity also showed increased DNA synthesis over those with low PI intensity. These results independently confirm, as expected, that PI staining intensity faithfully reports DNA content.

AD neuropathology strongly affects the prefrontal cortex. We therefore first interrogated the DNA content of pathologically confirmed prefrontal cortices (N = 32), using previously described methodologies for DCV analyses in neuronal nuclei (*Figure 1B*) (*Westra et al., 2010*). In control experiments, AD cerebellar nuclei showed DNA content profiles similar to lymphocytes, characterized by histograms with sharp peaks and narrow bases (*Figure 2C,E*). By comparison, AD cortices displayed high variability characterized by right hand shoulders (*Figure 2D,E* [AD-7]), large right hand peak shifts (*Figure 2D,E* [AD-6]), and wide bases. Of the AD cortex samples examined, greater than 90% displayed a net DNA content increase, averaging approximately 8% gain over human lymphocyte controls (*Figure 2F–H*). Notably, the AD cortex displayed increases beyond those observed in non-diseased cortices, with an average gain of approximately 6% over age and sex-matched samples (*Figure 2F–H*). AD cortices also displayed an increased coefficient of variation over non-diseased cortical nuclei and all cerebellar nuclei (*Figure 2I*) as well as a consistent skewed distribution compared to AD cerebellum (*Figure 2—figure supplement 1A*). In addition, we examined 14 paired sets of AD cortex and cerebellum from the same individual (*Figure 3A*) and 12 paired sets from non-diseased individuals (*Figure 3B*). Each AD cortex showed unique cortical histograms and increases in total cortical DNA compared to the cerebellum. In AD and non-diseased brain samples, DNA content changes did not correlate with age, Braak score, or postmortem index (*Figure 2—figure supplement 1B–H*), and DNA content was independent of nuclear size (*Figure 2—figure supplement 2*) (*Westra et al., 2010*). Importantly, the increase observed was significantly less than a 4N tetraploid genome (~12,800 Mb), yet significantly more than what would be expected from a hypersomy of even the largest chromosome (chr1: ~250 Mb or 3.9%). These results indicate that neuroanatomically selective increases in DNA content represent a distinct, reproducible, and prevalent characteristic of the AD brain.

## Neurons produce DNA content increases

In the non-diseased cortex, neurons were identified as the predominant cell type contributing to increased DCV (*Westra et al., 2010*). To determine whether neurons were also responsible for increased DCV in sporadic AD, nuclei were immunolabeled for the neuronal nuclear antigen, NeuN, and flow cytometry was used to analyze DCV in NeuN-positive and NeuN-negative nuclei (*Figure 4A–C*). NeuN-positive AD cortical neurons showed right hand DNA content shifts (*Figure 4D*) and increased DCV, displaying ~9% gain over that observed in NeuN-negative nuclei (*Figure 4E*). A comparison of NeuN-positive nuclei in paired samples from the same brain also showed significantly increased DNA content in cortical neurons over cerebellar neurons (*Figure 4F*). These distinctions do not rule out AD-specific effects on DCV from non-neuronal cells, but do implicate neurons as the major cellular locus for increased DCV.

## *APP* copy number variably increases in small cell cohorts

The DNA content increases in sporadic AD neurons raised the question of what genomic loci may be specifically increased in AD. The most highly validated gene associated with AD is *APP*, wherein constitutive gains have previously been linked to AD through familial AD and DS (*Prasher et al., 1998*; *Rovelet-Lecrux et al., 2006*; *Jonsson et al., 2012*; *McNaughton et al., 2012*). However, prior large-scale analyses in peripheral, non-neural tissues from sporadic AD cases have not reported *APP* copy

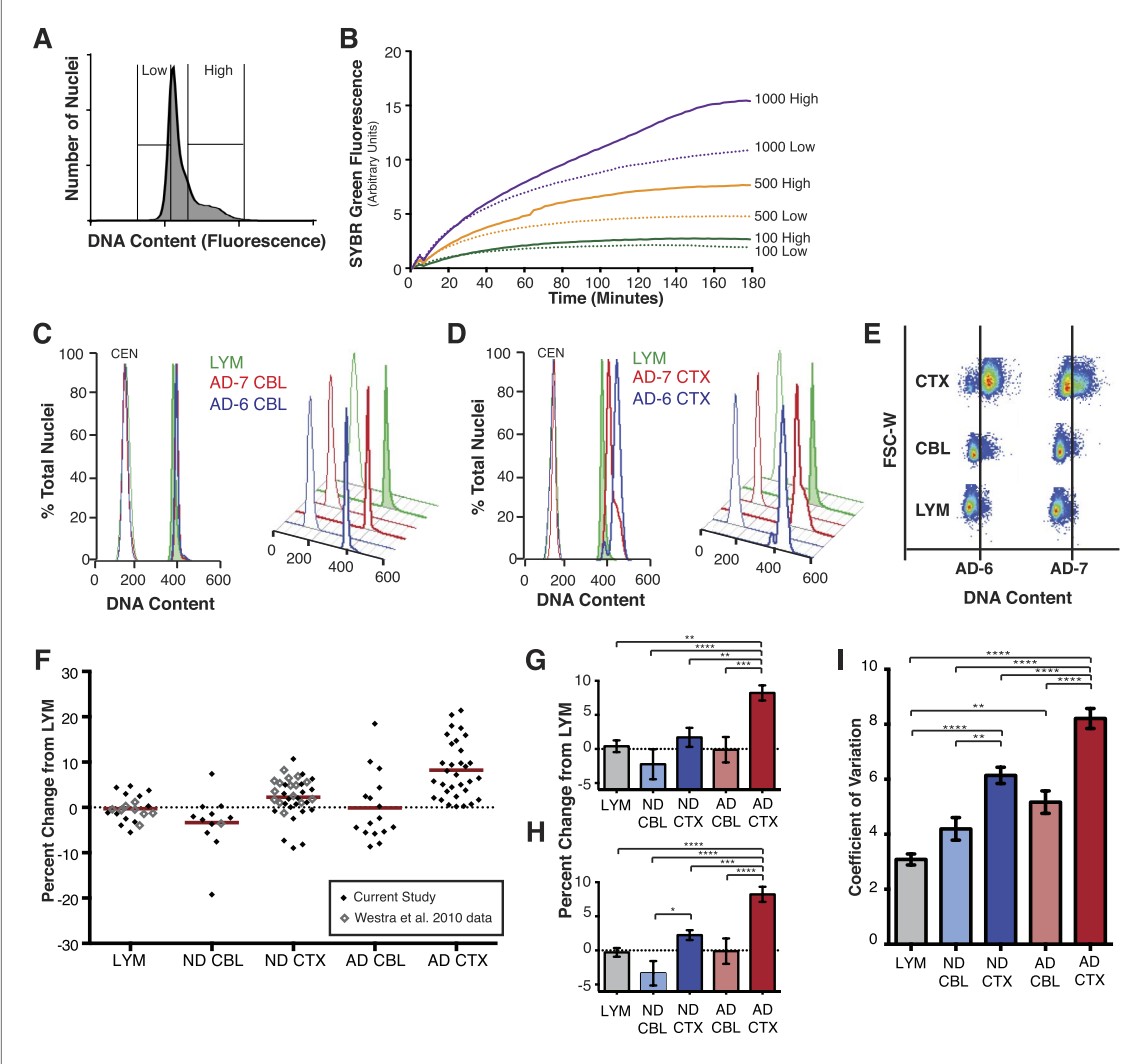

**Figure 2**. AD cortical nuclei show increased DNA content variation (DCV) by flow cytometry. (**A**) Histogram displaying gating parameters used in sorting 'high' and 'low' DNA content populations for validation of DNA content. (**B**) Validation of DNA content analyses using semi-quantitative MDA whole-genome amplification (WGA) on 'high' and 'low' DNA content populations of 1000, 500, and 100 nuclei. (**C** and **D**) Representative DNA content histograms for lymphocytes (LYM), AD cerebellum (CBL), and AD prefrontal cortex (CTX). Each colored histogram represents a separate sample in each set; CTX and CBL samples are from paired brains. Chicken erythrocyte nuclei (CEN) were used as internal calibration controls. (**E**) Representative orthogonal view of DNA content vs forward scatter width (FSC-W). For each brain sample, the area to the right of the vertical line indicates a DNA content increase of the population of nuclei. AD-6 CTX is a representative right-hand peak shift and AD-7 is a representative right-hand shoulder (see *Figure 3A* for more examples). (**F**) DNA content changes for all human LYM, ND, and AD brain samples examined (AD CTX N = 32, AD CBL N = 16, LYM N = 15 [20 meta analysis], ND CTX = 21 [36 meta analysis], ND CBL = 11 [12 meta analysis]). Red bars denote average for each group relative to lymphocytes. Averages are as follows (including metadata from *Westra et al. (2010)*): AD CTX 8.219%; AD CBL −0.1104%; LYM −0.2915%; ND CTX 2.239%; ND CBL −3.358%. (**G**) DNA content changes of the current study (AVOVA p < 0.0001). (**H**) DNA content changes of the current study combined with metadata from *Westra et al. (2010)* (ANOVA p < 0.0001). (**I**) Comparison of mean coefficient of variation (CV statistic from FlowJo of the population, included metadata from *Westra et al., 2010*) demonstrates that there is an average increase in the variation of AD samples (ANOVA p < 0.0001). *p = 0.05, **p = 0.01, ***p = 0.001, ****p < 0.0001, See *Figure 2—source data 1* for exact p values. See *Figure 2—figure supplement 1* for age, PMI and Braak score correlations. See *Figure 2—figure supplement 2* for control of nuclear size analysis.

The following source data and figure supplements are available for figure 2:

**Source data 1**. DNA Index (DI) and percent change values and statistics.

**Figure supplement 1**. DNA content shows no correlation with age or post-mortem index (PMI).

**Figure supplement 2**. Analysis of nuclear size and DNA content.

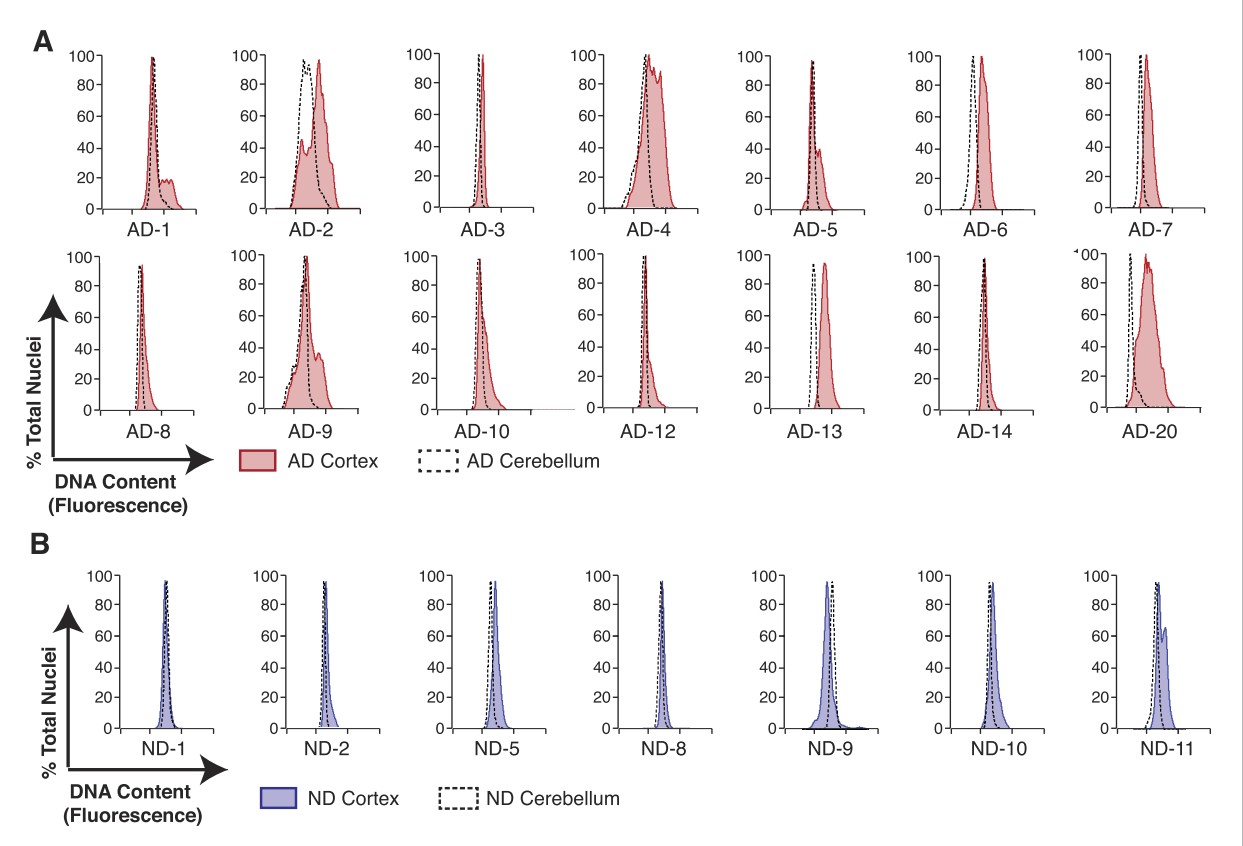

**Figure 3**. Pairwise DNA content analyses in AD cortical nuclei vs AD cerebellum. (**A**) Pairwise analysis of overlaid DNA content histograms (CTX = solid red, CBL = black dashed lines) in the same AD individual (each graph represents a unique AD individual). (**B**) Pairwise analysis of overlaid DNA content histograms (CTX = solid blue, CBL = black dashed lines) in the same ND individual.

number changes (*Bertram and Tanzi, 2009*; *Harold et al., 2009*). The normal existence of genomic mosaicism might produce 'hot-spots' of cells with increased *APP* copy number, and we therefore interrogated small cohorts of brain nuclei from paired cortex and cerebellum. Three of six AD samples displayed significant increases in *APP* copy number (3.9–6.3 copies compared to reference genes) over paired cerebellar samples (*Figure 5A* (AD-1, AD-3, AD-6)). Similar *APP* amplification was not detected in non-diseased brains (*Figure 5B*), and DS controls along with reference gene assessments demonstrated expected results (*Figure 5—figure supplement 1A,B*). Notably, AD samples displayed substantial variability in *APP* copies (*Figure 5C*), revealing inherent limitations in precisely quantifying copy numbers in small, genomically mosaic cell populations and highlighting the necessity for single-cell analyses.

### *APP* copy number increases in AD are not due to trisomy 21

*APP* gene dosage in familial AD and DS has driven hypotheses connecting AD pathogenesis with increased incidence of trisomy 21 (*Heston and Mastri, 1977*; *Potter, 1991*; *Geller and Potter, 1999*). We therefore examined AD cortical nuclei using a highly liberal protocol for calling aneusomies whereby borderline FISH profiles suggestive of aneusomy were always included in quantitative assessments, in an effort to detect possible differences between AD and control brains. Importantly, all analyses were conducted blind to the identity of samples, an approach made possible by interrogating purified nuclei rather than cells or tissues that themselves show identifying increases of plaques and tangles in AD. Nuclei were double-labeled with a commercially available chromosome 21 q-arm 'whole chromosome paint' (WCP) and a regional 'point' probe for 21q22.13-q22.2 (220 Kb) (*Figure 5D–I*). The ability of this technique to detect aneuploidy was validated using interphase nuclei from a human trisomy 21 brain (*Svendsen et al., 1998*) revealing three nuclear signals (*Figure 5D,E*) (see also (*Rehen et al., 2005*)). Three separate, blinded observers counted each sample. Despite using liberal

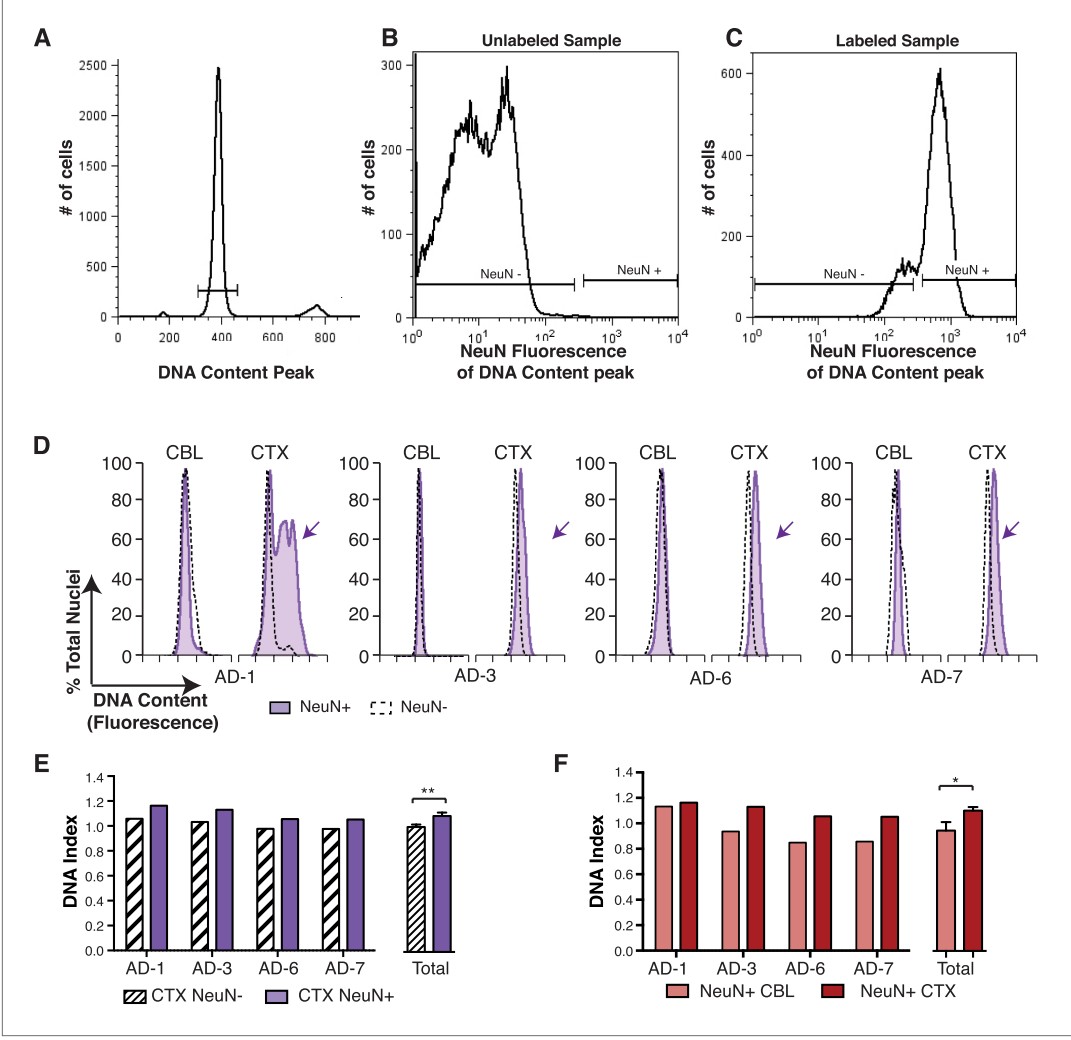

**Figure 4**. DNA content increases observed in AD cortical nuclei are attributable to neurons. (**A–C**) The gating procedure used for NeuN-positive flow cytometry analysis. (**A**) DNA content peak for identified nuclei. (**B**) A sample of unlabeled neuronal nuclei that display no NeuN-positive signal. (**C**) Selection for NeuN-positive nuclei for downstream analysis. (**D**) DNA content histograms of four AD samples displaying NeuN-positive nuclei (solid purple) vs NeuN-negative nuclei (black dashed line). NeuN-positive populations display distinct cortical histograms with prominent right-shifted peaks (arrows). (**E**) Comparison of DNA index (DI) increases from NeuN-positive nuclei (solid purple) vs NeuN-negative nuclei (black dashes) from AD CTX samples. NeuN-positive nuclei (DI = 1.10) showed an average gain of 9% over NeuN-negative nuclei (DI = 1.01), \*\*p = 0.0011. (**F**) Comparison of DNA content in NeuN-positive nuclei from AD CTX (DI = 1.10) (red) vs AD CBL (DI = 0.94) (pink) from the same individual; CTX nuclei displayed a 15.6% gain over CBL nuclei, \*p = 0.0335. Statistics are paired two-tailed *t*-test. Bars indicate ± SEM.

counting criteria, no statistically significant changes in chromosome 21 aneuploidy rates, including trisomies, between AD (N = 4974 nuclei, N = 9 brains) and non-diseased brains (N = 2576 nuclei, N = 5 brains) were observed (*Figure 5J*). Comparably high levels of aneuploidy have been reported for chromosome 21 displaying no difference between AD and non-diseased hippocampal cells (*Thomas and Fenech, 2008*). Thus, results from dual probe FISH analyses do not support increased trisomy 21 in AD but are consistent with alternative CNV mechanisms that could produce *APP* copy number values exceeding 3, as was observed in 75-genome qPCR analyses (*Figure 5*).

### *APP* copy number is increased and enriched in single AD neurons

Mosaic single-cell increases in *APP* copy number could explain variations observed in small cohort qPCR data. To assess *APP* copy number in single neurons, an optimized microfluidic protocol to

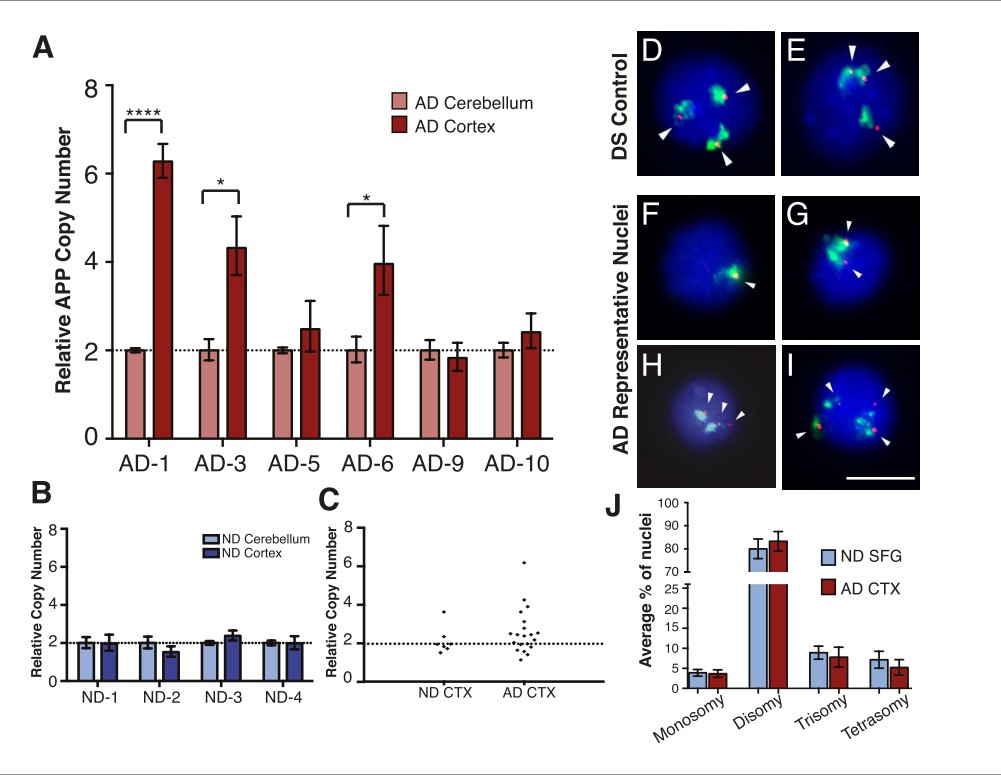

**Figure 5**. Mosaic amplification of the *APP* locus in small cohorts of AD cortical neurons unrelated to trisomy 21. (**A**) Comparison of relative copy number of *APP* in CBL and CTX fractions from six AD brains. *APP* locus-specific amplification was determined relative to reference gene *SEMA4A*; paired CBL nuclei were used as a calibrator sample for each brain, normalized to 2.00 for a diploid cell. Differences in ΔΔCt ± SEM of *APP* in the cortex vs cerebellum were assessed in each individual using an unpaired, two-tailed t-test (****p = 0.0001, *p = 0.0165, *p = 0.0489) (**B**) Comparison of relative copy number of *APP* in CBL and CTX fractions from 4 non-diseased brains. (**C**) Average relative copy number in non-diseased vs AD brains. Control genes and DS individuals were also examined (***Figure 5—figure supplement 1***). (**D–J**) FISH strategy of chromosome 21 counting through simultaneous labeling using chr 21 q arm 'whole' chromosome paint (WCP, green) and chr 21 regional FISH probe for 21q22.13-q22.2 (red) (see ***Figure 5—source data 1*** for raw counts). (**D** and **E**) The ability to detect aneuploidy was validated using interphase nuclei from a human trisomy 21 brain, where three regional spots (red, encompassing the *APP* gene) were seen, despite WCP spatial variation (see also ***Rehen et al. (2005)***). (**F–I**) Chromosome 21 aneusomy was examined in prefrontal cortical nuclei. Examples of chr 21 (**F**) monosomy, (**G**) disomy, (**H**) trisomy, and (**I**) tetrasomy (please note tetrasomy is not an example of aneuploidy). (**J**) Quantification of individual FISH signals showed no significant differences in monosomy, disomy, trisomy, or tetrasomy. 5 control brains and 9 AD brains were used. At least 450 nuclei were quantified per brain sample. Scale bar = 10 um. 4974 total nuclei examined.

The following source data and figure supplement are available for figure 5:

**Source data 1**. Raw dual point-paint probe FISH counts.

**Figure supplement 1**. Controls for small population qPCR.

assess genomic copy number via qPCR using a Biomark HD 48.48 Dynamic Array integrated fluidic circuit (IFC) (Fluidigm, South San Francisco) was adapted from gene expression protocols. Nuclei were isolated and interrogated at two reference genes (see Materials and methods) and two *APP* exons that flanked the majority of the coding sequence at the 5' and 3' ends of *APP* (exons 3 and 14). Samples were run in triplicate and assays in sextuplicate, which generated high numbers of replicate data points allowing for improved copy number resolution to an extent not possible by conventional qPCR (***Weaver et al., 2010***; ***Whale et al., 2012***). A total of 154 neuronal nuclei were individually examined from three AD and three non-diseased brains, within which nuclei from the cerebellum and prefrontal cortex were separately analyzed. In single AD cortical nuclei, significant increases in *APP*

copy number, ranging up to 12 copies in a single nucleus (*Figure 6A,B*), were observed with a high concordance rate between exons (*Figure 6—figure supplement 1*). AD cortical nuclei displayed increased average copy numbers (~3.8–4 copies) over control samples (~1.7–2.2 copies) (*Figure 6A,B*), with increased frequencies of high copy number nuclei (six or more copies) (*Figures 6C and 5E*, red lines) primarily occurring in prefrontal cortex samples (*Figure 6C–F*). The AD cortex showed an approximately fourfold increase over the non-diseased cortex of nuclei with greater than 2 *APP* copies (55–66% vs 12–15%) (*Figure 6G,I*). In addition, nuclei with fewer than 2 *APP* copies were also identified. In neuronal nuclei with greater than two copies (*Figure 6G,I*, gold bars), AD cortical nuclei showed statistically significant increases (~5 copies for both exons) over the other samples (~3 copies for both exons) (*Figure 6H,J*). These data identify mosaic, neuroanatomically enriched and disease-associated increases in *APP* copy number in single, sporadic AD neurons.

## *APP* amplified loci can be visualized by PNA-FISH

The identification of high *APP* copy number nuclei by single-cell qPCR suggested the possibility that amplified loci could produce detectable FISH signals. In chromosome 21 non-quantitative regional point probe FISH experiments, we occasionally observed variable puncta sizes (*Figure 5H,I*) (*Rehen et al., 2005*) that were initially dismissed as technical hybridization variability but which might potentially indicate sub-chromosomal variation of loci containing target sequences. However, conventional point-probe FISH analysis, while useful for assessments of aneusomy or rearrangements, cannot quantitatively evaluate copy number changes occurring as contiguous copies in close proximity to one another. Moreover, other conventional techniques for identifying copy number in single cells, such as single cell sequencing, rely heavily on genome amplification, which may introduce bias or mask CNVs. We therefore developed a detection assay based upon PNA chemistry. PNA probes hybridize with single base discrimination and have been used to quantify short repeat sequences such as those found on telomeres (*Buchardt et al., 1993*; *Lansdorp et al., 1996*). This raised the possibility that amplified copies of *APP* could be identified using multiple PNA probes simultaneously hybridized to unique gene sequences, provided that amplification occurred in a physically constrained locus rather than being dispersed throughout the genome.

Nine PNA probes of 12–18 residues, each conjugated to a single Alexa-488 fluor, were designed against multiple sites of the same *APP* exons examined by single-cell qPCR, and validated both in silico and by blotting for specificity and linear quantitative behaviors (*Figure 7A*, *Figure 7—figure supplement 1A–C*). Relative probe sequence locations were also designed to avoid fluorochrome proximity quenching. Single PNA probes did not produce a detectable signal in any samples using standard fluorescence microscopy used for aneusomy FISH (*Rehen et al., 2001*, *2005*), contrasting with clearly detectable telomeric signals, which led to evaluation of more sensitive microscopic techniques including confocal, deconvolution (*Westra et al., 2010*), and ultimately super resolution structured illumination microscopy (N-SIM, Nikon) (*Gustafsson, 2000*), which all failed to detect possible signals in AD nuclei. However, hybridization of increasing numbers of distinct PNA probes combined with N-SIM visualization resulted in the empirical determination of a threshold that identified one, and rarely two, punctate signals that were much more frequent in AD nuclei (*Figure 7B*), and with a frequency consistent with single-cell qPCR data (*Figure 6*). AD neuronal nuclei showed single purely green puncta (*Figure 7B*, #1 arrow, *Video 1–6*) and rarer two puncta signals (*Figure 7B*, #3 arrow, *Video 6*), all of which could be differentiated from lipofuscin puncta that fluoresced in all channels (*Figure 7B*, #2 arrow, *Video 5*). Internal positive control signals identified telomeres (*Figure 7*, red fluorescent puncta). Use of 3D-SIM enabled acquisition and analysis of super resolution projections of individual green puncta (*Figure 7B*) revealing a range of morphologies and intensities (*Figure 7B,D*, *Figure 7—figure supplement 1D,E*) consistent with varied *APP* copy number and possibly distinct, intrachromosomal genomic organization. This threshold detection using multiple PNA probes targeting *APP* visualized green puncta in 56% of AD neuronal nuclei compared to 22% in non-diseased and 14% in DS (*Figure 7C*). There was no evidence of single-copy detection in any sample, contrasting with standard FISH data (*Figure 5D–I*), despite always showing telomeric signals. Compared to single-cell qPCR data, which reports both endogenous alleles in addition to amplification, copy numbers at a *single* locus below ~2 were not visualized with PNA-FISH, allowing focused analyses on positive profiles resulting in fluorescence intensity plots (*Figure 7D* and *Figure 7—figure supplement 1D*) that were highly reminiscent of *APP* copy number plots produced by single-cell qPCR (*Figure 6*).

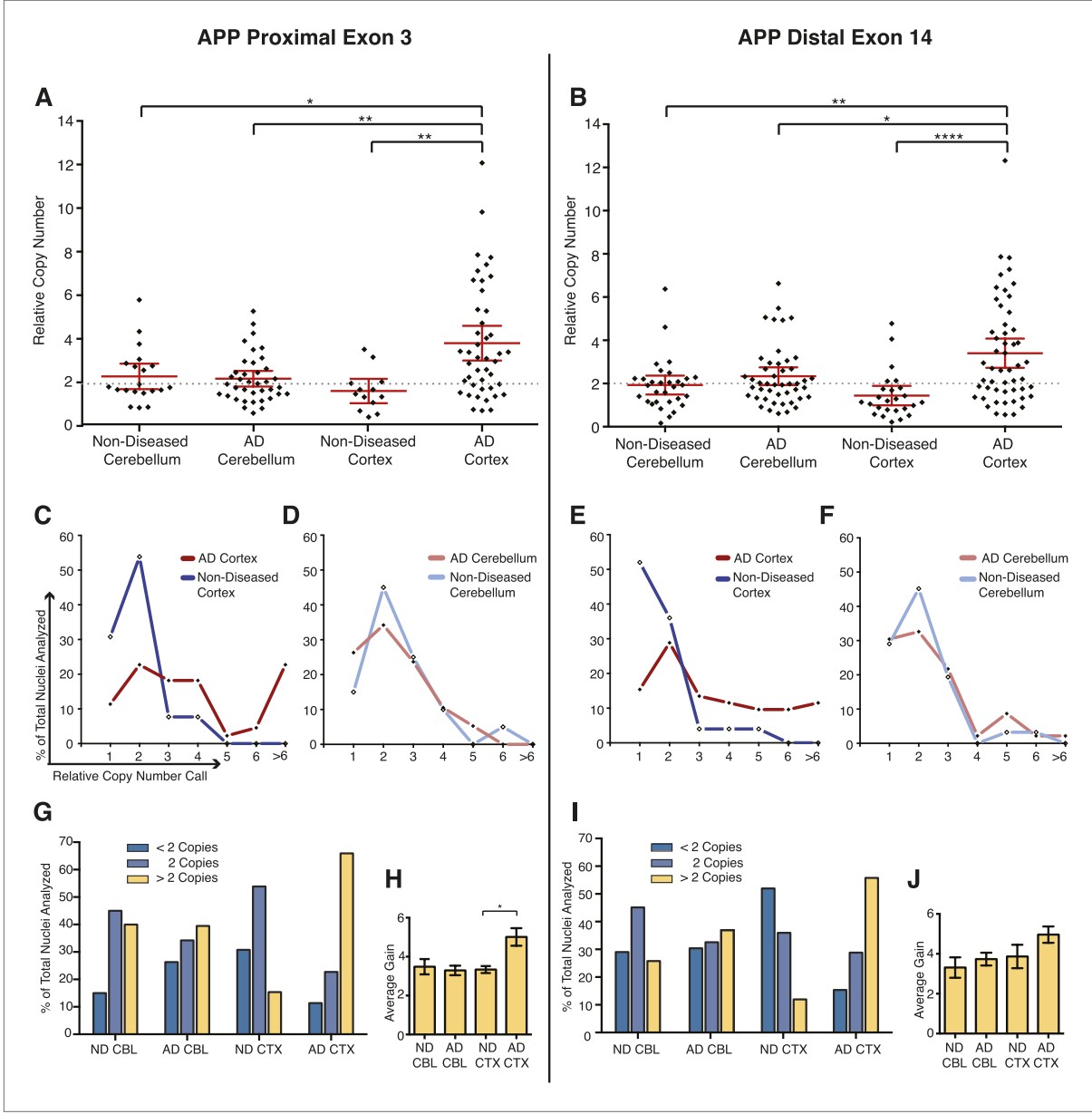

**Figure 6**. Mosaic *APP* locus amplification in single neurons from AD brains. (**A**) Single nuclei relative copy numbers for exon 3 of *APP* from non-diseased (ND) CBL, ND CTX, AD CBL, and AD CTX; each black diamond represents one neuron. For each group, the mean is displayed in red and bars represent 95% confidence intervals. AD CTX showed a mean *APP* copy number of 3.80; this is significantly higher than AD CBL (2.23), ND CTX (1.60), and ND CBL (2.28). *p = 0.0147, **p = 0.0015, **p < 0.0012, ANOVA p < 0.0001 (see *Figure 6—source data 1* for raw numbers and statistics). (**B**) Single nuclei relative copy numbers for exon 14 of *APP*, similar to (**A**). The two exons showed a high concordance (*Figure 6—figure supplement 1*) where the AD CTX showed a mean *APP* copy number of 3.40 while the AD CBL (2.34), ND CTX (1.44), and ND CBL (1.92) remained closer to 2 copies. *p = 0.0163, **p = 0.0016, ****p < 0.0001, ANOVA p < 0.0001. (**C**–**F**) Distribution of copy number calls for exon 3 (**C** and **D**) and exon 14 (**E** and **F**) binned by relative copy number. The AD CTX for both exons displayed unique distributions, with more nuclei falling into the high copy number bins. (**G** and **I**) Distribution of nuclei with copy numbers less than, equal to, and greater than two copies. (**H** and **J**) Average copy number increases in nuclei binned with greater than two copies (gold columns in **G**) (AD CTX: Exon 3 = 5.01, Exon 14 = 4.96, *p = 0.0361). All statistics represent an ANOVA with a Tukey's multiple comparison test. Bars indicate ± SEM.

The following source data and figure supplement are available for figure 6:

**Source data 1**. Single Cell qPCR Data and Statistics.
**Figure supplement 1**. Concordance of *APP* exon 3 and 14 from single cell qPCR.

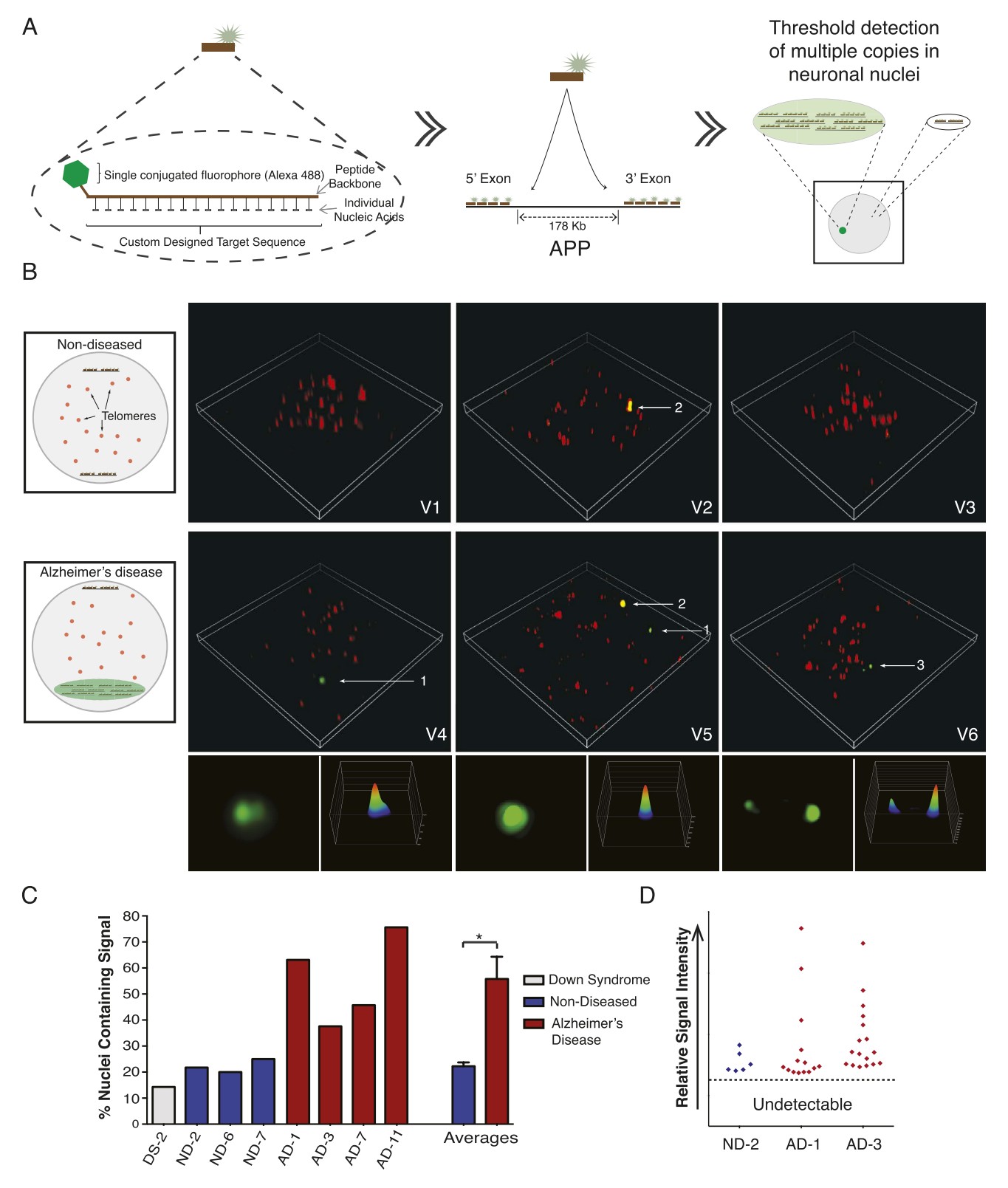

**Figure 7**. Visualization of *APP* copy number increases in neuronal nuclei from AD brain samples. (**A**) Peptide nucleic acid probes (PNA) were developed against nine separate sites on *APP* (4 sites within exon 3 and 5 sites within exon 14). Each PNA probe consists of a peptide backbone conjugated to a single fluorophore, with separately conjugated nucleotides, substantially increasing specificity (***Lansdorp et al., 1996***). Single copies of *APP* are not

*Figure 7. Continued on next page*

*Figure 7. Continued*

detectable because of fluorophore detection limits. Detection of increased copy number by PNA probes can be visualized as copies of *APP* increase (*Figure 7—figure supplement 1B,C*). Positive internal controls using PNA probes directed against telomere sequences were simultaneously hybridized. (**B**) Visualization of copy number increases in neuronal nuclei. Green puncta (arrow 1, insets) indicate visualized *APP* increases. Telomere labeling (red puncta) was present in all nuclei, demonstrating probe accessibility and template fidelity. Lipofuscin (arrow 2, orange puncta) was detected in nuclei, visualized by extensive fluorescence signal in all channels, but was eliminated from quantifications. Limited nuclei displayed two green puncta (arrow 3). V1-6 Refers to the supplemental videos where 3-D projections can be visualized. (**C**) Graphic representation of non-diseased (blue) and DS (grey) brains displayed limited numbers of threshold-detected increases in *APP* (*Figure 7—source data 1*). AD (red) brains displayed significant and consistent threshold-detected increases in *APP*. (**D**) Individual threshold-detected *APP* increases were quantified and plotted on a relative intensity scale (blue diamonds: non-diseased, red diamonds: AD). Dotted line represents the threshold below which *APP* copy number was undetectable, only limited puncta were identified in non-diseased nuclei. Bars indicate ± SEM, *p < 0.05.

The following source data and figure supplement are available for figure 7:

**Source data 1**. Data and statistics for PNA-FISH counts.

**Figure supplement 1**. PNA FISH controls.

## Discussion

Our results support a mechanism for the development of sporadic AD whereby the validated familial pathogenic *APP* gene is mosaically amplified in neurons. Mosaicism was documented based on multiple independent criteria: increased DCV in 90% of examined AD brains, varied and increased *APP* copy number in small cohorts, increased *APP* CNVs in single neurons identified by single-cell qPCR, and *APP* copy number increases in single nuclei visualized by PNA-FISH. These data are consistent with our inability to substantiate increases in trisomy 21 in cells of the sporadic AD brain. The concomitant increase in DCV suggests effects on other loci that are also altered in AD, the most obvious of which would be copy gains, but which could also involve losses based upon the presence of increased DCV range observed in AD; such loci might contribute to the progeric presentation of sporadic AD and possibly familial forms of AD that still require decades for disease to manifest. Our working hypothesis ties the currently accepted pathogenicity of *APP* gene dosage, established by familial AD and DS, to sporadic AD through a mechanism of somatic, mosaically increased *APP* copy number in some neurons. Our data do not exclude the possibility that the changes observed are a downstream effect of causative factors in the disease. However, if mosaic genomic changes are downstream of disease onset, *APP* copy number changes would likely play a significant role in disease progression.

Advances in single-cell genomic sequencing had initially suggested its use for identifying *APP* CNVs in single AD neurons. However, existent technology is limited to published sequence resolution between ~2-5 Mb and ~0.025X genome coverage (*Evrony et al., 2012*; *Gole et al., 2013*; *McConnell et al., 2013*; *Cai et al., 2014*). Considering the <0.3 Mb size of the *APP* locus, single cell sequencing would be incapable of identifying all but the rarest *APP* CNVs observed here. In addition, whole genome single cell sequencing is limited by throughput, demonstrated by the low number of neurons reported in this field, and is complicated further by variable results (Cai et al. N = 82 (QC neurons), Evrony et al. N = 6, Gole et al. N = 6, McConnell et al. N = 110). Notably, DNA losses predominated in both *Cai et al. (2014)* and *McConnell et al. (2013)* however, *Gole et al. (2013)* observed that two thirds of somatic CNVs were gains, consistent with increases in DNA content reported previously, and all three studies support a *range* of DCV amongst neurons, although none of these prior studies assessed AD neurons. While future advances will improve sequence resolution, throughput, and amplification fidelity in single cells, the distinct single cell strategy employed here allowed copy number assessment of a single targeted gene, *APP*, an approach which may be generalizable to other loci and diseases. Compared to the prior single-neuron reports, 320 single, neuronal nuclei were assessed here for *APP* CNVs wherein single cell qPCR data markedly resembled PNA-FISH data. Notably, these single cell techniques also possess limitations. For example, single cell qPCR requires normalization to single copy reference genes and control populations, which presents unique difficulties in assessing mosaic neuronal populations. For example, changes in copy number could reflect changes in both the reference and target gene, which might result in artifactual increases (or decreases) in a target gene. Similarly, assessment of copy number from an amplified template could also produce artifactual changes, and thus results require independent techniques for verification, especially since

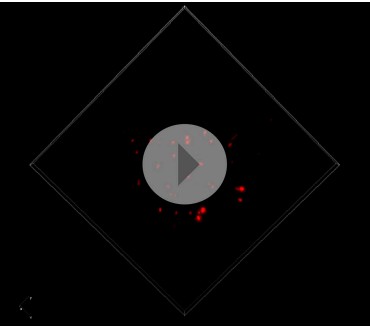

**Video 1**. PNA-Fish analysis of *APP* in nuclei from non-diseased cortical neurons. Video of 3-D projection from *Figure 7B*, V1. Green puncta indicating *APP* increases were infrequently visualized in non-diseased brains. Red puncta indicate telomere labeling with separate telomere-specific PNA probes and were visualized as a positive control for PNA hybridization.

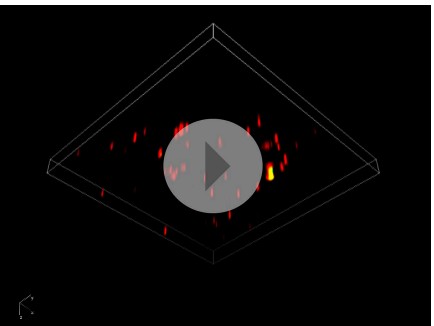

**Video 2**. PNA-Fish analysis of *APP* in nuclei from non-diseased cortical neurons with lipofuscin. Video of 3-D projection from *Figure 7B*, V2. Green puncta indicating *APP* increases were infrequently visualized in non-diseased brains. Lipofuscin (orange puncta), visualized by extensive fluorescence signal in all channels, was detected in some nuclei, but was excluded from analysis. Red puncta indicate telomere labeling with separate telomere-specific PNA probes and were visualized as a positive control for PNA hybridization.

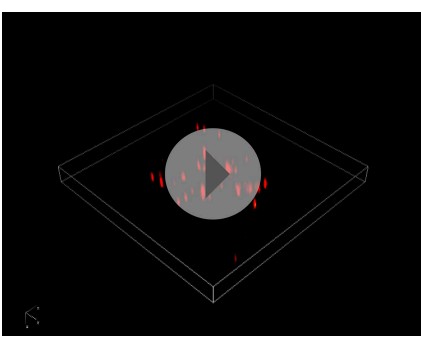

**Video 3**. PNA-Fish analysis of *APP* in nuclei from non-diseased cortical neurons. Video of 3-D projection from *Figure 7B*, V3. Green puncta indicating *APP* increases were infrequently visualized in non-diseased brains. Red puncta indicate telomere labeling with separate telomere-specific PNA probes and were visualized as a positive control for PNA hybridization.

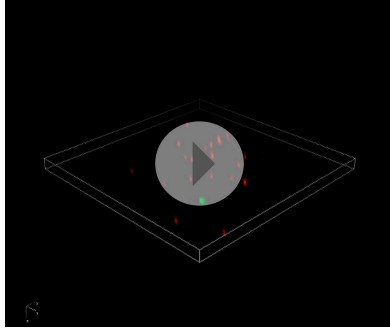

**Video 4**. PNA-Fish analysis of *APP* in nuclei from AD cortical neurons. Video of 3-D projection from *Figure 7B*, V4. Green puncta indicate visualized *APP* increases. Red puncta indicate telomere labeling with separate telomere-specific PNA probes and were visualized as a positive control for PNA hybridization.

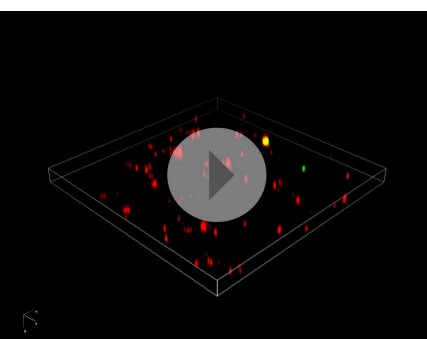

**Video 5**. PNA-Fish analysis of *APP* in nuclei from AD cortical neurons with lipofuscin. Video of 3-D projection from *Figure 7B*, V5. Green puncta indicate visualized *APP* increases. Lipofuscin (orange puncta), visualized by extensive fluorescence signal in all channels, was detected in some nuclei, but was excluded from analysis. Red puncta indicate telomere labeling with separate telomere-specific PNA probes and were visualized as a positive control for PNA hybridization.

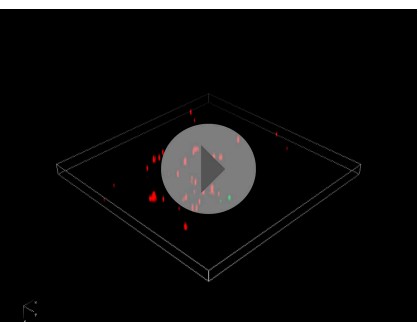

**Video 6**. PNA-Fish analysis of *APP* in nuclei from AD cortical neurons. Video of 3-D projection from *Figure 7B*, V6. Green puncta indicate visualized *APP* increases. Limited nuclei in AD displayed two green puncta. Red puncta indicate telomere labeling with separate telomere-specific PNA probes and were visualized as a positive control for PNA hybridization.

the original template is consumed in the single cell reaction and cannot be further assessed. The use of PNA-FISH does not require any amplification or normalization, and yet PNA-FISH and single-cell qPCR produced a highly similar distribution of APP copy number amplifications. The employed techniques were not capable of assessing intact neurons or their histological organization, which will require technological modifications. Similarly, assessments of genomic and expression data in a single neuron also await technological advancements. Finally, these techniques are currently capable of interrogating only parts of the entire APP locus; therefore, at this time the precise structure of APP CNVs and the mechanisms leading to their existence are unknown.

Our data have bearing on at least 3 AD hypotheses. First, the prevailing amyloid hypothesis in AD posits that Aβ deposition drives AD (*Hardy and Selkoe, 2002*). Increased incidences of amyloid senile plaques are observed in all forms of AD and appear to be directly linked to *APP* dosage in DS and familial AD. The mosaic increases in neuronal *APP* copy number reported here provide an explanation for the universal presence of Aβ senile plaques in sporadic forms—indeed, all forms—of AD despite an absence of constitutive copy number gain. The presence of rare neurons with *APP* amplifications in non-diseased brains also provides an explanation for the observed presence of senile plaques in otherwise normal, aged brains (*Gibson, 1983*; *Cras et al., 1991*; *Mackenzie et al., 1996*), consistent with the ages of non-diseased brains examined here, which exceeded 75 years. Along with the occurrence and augmentation of DCV, *APP* amplification in AD is consistent with dysregulation of normally occurring processes in the etiology of AD pathogenesis. A second hypothesis proposes increased *APP* dosage through trisomy 21 based largely upon the neuropathology of DS (*Heston and Mastri, 1977*; *Potter, 1991*; *Geller and Potter, 1999*). However, dual chromosome point-paint FISH for chromosome 21 using liberal calling criteria on ~5000 AD brain cells (N = 14 brains) showed no relationship between total mosaic aneusomies and AD. Moreover, the lack of observed increases of trisomy 21 in sporadic AD are consistent with prior studies in both peripheral non-brain and intact brain tissues, which reported *APP* levels approximating 2N (*Podlisny et al., 1987*; *St George-Hyslop et al., 1987*; *Tanzi et al., 1987*; *Bertram et al., 2010*), and is further consistent with mosaic CNVs observed here. The purposeful use of liberal counting criteria resulted in generally higher percentages of aneusomies than reported previously (*Rehen et al., 2005*; *Iourov et al., 2009*), however the increases were observed in both AD and non-diseased samples without linkage to AD. Our results do not eliminate roles for mosaic aneuploidy in AD, but do not support trisomy 21 as a specific mechanism for increasing *APP* copy number in sporadic AD. A third hypothesis is that abnormal cell cycle reentry in post-mitotic neurons contributes to AD (*Yang et al., 2001*; *Herrup and Arendt, 2002*; *Kruman II et al., 2004*; *Copani et al., 2006*; *Mosch et al., 2007*; *Herrup, 2012*). Our DCV analyses detected maximum gains of ~21% and average gains of ~8.2% over lymphocytes, representing a fraction of the 100% increase expected in a 4N cell. The subgenomic increases observed here are also consistent with the reported absence of adult neurogenesis in the normal cerebral cortex (*Rakic, 2002*; *Bhardwaj et al., 2006*), and could also be relevant to reports of nucleotide incorporation in adult cortical neurons (*Gould et al., 1999*). Concepts involving changes in DNA synthesis in AD without cell-cycle progression or neurogenesis could be rectified through subgenomic DNA synthesis in post-mitotic neurons that might involve DNA synthetic proteins reported in AD (*Copani et al., 2002*, *2006*).

The observed genomic alterations in AD could occur through both developmental as well as aging processes as reported for at least one form of mosaicism, aneuploidies (*Bushman and Chun, 2013*) that have been linked to caspase-mediated cell death (*Peterson et al., 2012*) and could have relevance to developmental *APP* functions (*Nikolaev et al., 2009*). Perhaps tellingly, these developmental cell death processes also involve DNA fragmentation and double strand breaks (*Blaschke et al., 1996*; *Staley et al., 1997*; *Blaschke et al., 1998*) that have recently been reported in neurons exposed to physiological and AD stimuli (*Suberbielle et al., 2013*). It is possible that somatic mosaicism may arise through dysfunction of normally operative DNA repair mechanisms that produce DCV in the normal brain, with augmentation in the diseased brain. We speculate that increased DCV and *APP* CNVs observed in AD may in part reflect neurons that have delayed or averted cell death, wherein somatic genomic changes could provide a survival advantage at a cost of altered neurophysiological functions.

The DNA content changes observed here showed not only disease relationships but also varied with brain region and cell type. It is important to note that the interrogated neurons are likely terminal in their fate: they will not divide further, contrasting with lymphocytes that, perhaps surprisingly, showed the least amount of DCV despite being a proliferating population. Unlike neurons, proliferating lymphocytes may be similar to stem and progenitor populations that can maintain germline

genomes to promote population expansion. By contrast, neurons are by definition post-mitotic, and therefore not subject to this possible requirement. We think it likely that many dividing cells that deviate significantly from the germline genome—and thus would be incapable of further division—are eliminated by cell death and would be removed from the population interrogated for DNA content. Support for this view includes reports on aneuploidies in neuroprogenitor cells, whereby the most extremely aneuploid forms appear to be eliminated by caspase-mediated cell death (*Blaschke et al., 1996*, *1998*; *Peterson et al., 2011*; *Peterson et al., 2012*). DNA content changes may therefore be contextual, enabling diverse functions in different cell types. The large differences observed even between brain regions and amongst neuronal nuclei would suggest the existence of distinct neuronal functions produced by DCV, and possibly specific processes or stimuli that can genomically alter neurons.

Diverse stimuli previously implicated as AD risk factors including age and trauma (*Sparks et al., 1990*; *Plassman et al., 2000*), might share common endpoints via influences on genomic mosaicism. The consequences of somatic *APP* amplification in AD may support functionality of other somatically altered genomic loci observed in single neurons that could contribute to the progeric presentation of AD as well as aspects of the disease itself. We further hypothesize that other sporadic or idiopathic brain diseases could arise through altered genomic mosaicism that includes somatic variations at both known and unknown pathogenic loci.

## Materials and methods

### Human tissue samples

All human tissue protocols were approved by the Scripps Office for the Protection of Research Subjects (SOPRS) at The Scripps Research Institute (TSRI) and conform to National Institutes of Health guidelines. Fresh-frozen brain tissue was provided by the NICHD Brain and Tissue Bank for Developmental Disorders at the University of Maryland, the University of California Alzheimer's Disease Research Center (UCI-ADRC), and the Institute for Memory Impairments and Neurological Disorders, the Johns Hopkins School of Medicine Alzheimer's Disease Research Center, and Dr Edward Koo at the University of California, San Diego. Lymphocytes from human peripheral blood were obtained from healthy donors at TSRI's Normal Blood Donor Services. Detailed information about the samples used can be found in the table below (*Table 1*).

### Flow cytometry (FCM) and fluorescence-activated cell sorting (FACS)

Human brain nuclei were isolated and prepared for FCM and FACS as previously described (*Westra et al., 2008*, *2010*). Isolated nuclei were fixed with 2% paraformaldehyde (or 70% ethanol for microfluidic qPCR), labeled with mouse anti-NeuN antibody (1:100) (Millipore, Germany) and Alexa Fluor 488 goat anti-mouse IgG secondary (1:250) (Life Technologies, Carlsbad, CA), and counterstained with propidium iodide (50 µg/ml) (Sigma, St. Louis, MO) in solution containing 50 µg/ml RNase A (Qiagen, Valencia, CA) and chicken erythrocyte nuclei (CEN) (Biosure, Grass Valley, CA). Electronically gated diploid neuronal nuclei, determined by PI fluorescence and immunolabeling, were analyzed and sorted either in bulk for standard qPCR and PNA-FISH or singly in 96-well plates for microfluidics-based qPCR. FCM and FACS were performed at the TSRI Flow Cytometry Core using a Becton Dickinson (BD Biosciences, San Jose, CA) LSRII and FACS-Aria II, respectively. *Post hoc* DNA content analyses were performed using FlowJo software (TreeStar Inc., Ashland, OR).

### DNA content assessment by whole genome amplification

For validation of DNA content analyses, nuclei were sorted by FACS in populations of 1000, 500, and 100 into 96-well plates according to relative DNA content (*Figure 2A*). Nuclei were denatured by multiple freeze–thaw cycles and potassium hydroxide, the solution was neutralized, and phi29 DNA polymerase (Illumina, San Diego, CA), dNTPs, SYBR green, and random hexamers (IDT, Coralville, IA) were added (*Gole et al., 2013*). The MDA reaction was performed at 30°C and SYBR Green intensity was recorded every 2 min on a QuantStudio RT-PCR 12K Flex (Life Technologies, Carlsbad, CA). 12 replicates were run and averaged for each group.

### Quantitative PCR primers

All qPCR was performed according to MIQE guidelines (*Bustin et al., 2009*). For standard qPCR, primers against the *APP* gene exon 14 and reference genes *SEMA4A*, *CCL18*, and *PCDH11X* were

**Table 1.** Human samples used in each experiment

| | Sex | Age | Paper code | Sample | Experiments | Post mortem interval | Braak score |
|---|---|---|---|---|---|---|---|
| Alzheimer's disease prefrontal cortex, N = 32 | F | 81 | | 1521 | D | 23 | VI |
| | F | 83 | | 1562 | D | 6 | U |
| | F | 74 | **AD-13** | **1866** | DT | U | U |
| | F | 79 | **AD-5** | **1868** | DST | U | U |
| | F | 82 | **AD-10** | **1875** | DST | U | U |
| | F | 83 | | 1893 | D | 9 | U |
| | F | 87 | | 1899 | D | 5 | VI |
| | F | 62 | **AD-9** | **1912** | DST | U | U |
| | F | 80 | **AD-8** | **1913** | DT | U | U |
| | F | 54 | **AD-4** | **1916** | DTB | U | U |
| | F | 72 | **AD-2** | **1921** | DTB | U | U |
| | F | 77 | | 2400 | D | 3.7 | V |
| | F | 80 | **AD-11** | 2500 | DP | 2.3 | VI |
| | F | 101 | | 50341 | D | 19 | V |
| | F | 91 | | 61788 | D | 11 | V |
| | F | 98 | | 62405 | D | 11 | V |
| | F | 89 | | 62439 | D | 11 | V |
| | F | 77 | | 62509 | D | 18 | VI |
| | M | 88 | **AD-1** | **102** | DSBP | 3 | IV |
| | M | 90 | | 268 | D | 91 | V |
| | M | 83 | | 736 | D | 27 | U |
| | M | 82 | | 1211 | D | 18 | U |
| | M | 79 | **AD-15** | **1252** | D | 9 | U |
| | M | 92 | | 1748 | D | 5.5 | V |
| | M | 85 | **AD-12** | **1861** | DT | U | U |
| | M | 85 | **AD-14** | **1870** | DT | U | U |
| | M | 82 | **AD-3** | **2401** | DSBP | 3 | VI |
| | M | 84 | **AD-6** | **2499** | DS | 3.4 | VI |
| | M | 63 | **AD-7** | **4199** | DP | 3 | VI |
| | M | 80 | | 13173 | D | 22 | IV |
| | M | 94 | | 30022 | D | 20 | V |
| | M | 91 | | 60987 | D | 22.5 | V |
| | **Mean** | **82.1** | | | | | |
| Alzheimer's disease cerebellum, N = 15 | F | 74 | **AD-13** | **1866** | D | U | U |
| | F | 79 | **AD-5** | **1868** | DS | U | U |
| | F | 82 | **AD-10** | **1875** | DS | U | U |
| | F | 80 | **AD-8** | **1913** | D | U | U |
| | F | 62 | **AD-9** | **1912** | DS | U | U |
| | F | 54 | **AD-4** | **1916** | DB | U | U |
| | F | 72 | **AD-2** | **1921** | DB | U | U |
| | M | 88 | **AD-1** | **102** | DSB | 3 | IV |
| | M | 79 | **AD-15** | **1252** | D | 9 | U |
| | M | 70 | | 1625 | D | 1 | U |
| | M | 85 | **AD-12** | **1861** | D | U | U |
| | M | 85 | **AD-14** | **1870** | D | U | U |

*Table 1. Continued on next page*

*Table 1. Continued*

|  | Sex | Age | Paper code | Sample | Experiments | Post mortem interval | Braak score |
|---|---|---|---|---|---|---|---|
|  | M | 82 | **AD-3** | **2401** | DSB | 3 | VI |
|  | M | 84 | **AD-6** | **2499** | DS | 3.4 | VI |
|  | M | 63 | **AD-7** | **4199** | D | 3 | VI |
|  | **Mean** | **75.9** |  |  |  |  |  |
| Non-diseased prefrontal cortex, N = 40 | F | 74 | **ND-2** | **1901** | DSBP | 2.3 | II |
|  | F | 74 | **ND-8** | **299** | D | 2.8 | II |
|  | F | 84 | **ND-3** | **703** | S | 5.8 | III |
|  | F | 53 | **ND-11** | **1379** | D | 15 | III |
|  | F | 73 |  | 713* | T | U | U |
|  | F | 95 |  | 60831 | D | 9 | II |
|  | F | 51 |  | 1568 | P | 22 | U |
|  | F | 17 |  | 1230 | P | 16 | U |
|  | F | 87 | **ND-10** | **1502** | D | 5 | II |
|  | F | 80 |  | 60728 | D | 13 | II |
|  | M | 79 | **ND-9** | **827*** | DT | U | U |
|  | M | 96 | **ND-1** | **1102** | DSB | 3.4 | II |
|  | M | 83 | **ND-5** | **2501** | DB | 1.7 | II |
|  | M | 95 | **ND-4** | **1301** | DS | 3.5 | I |
|  | M | 87 |  | **1471*** | T | U | U |
|  | F | 71 |  | **1571*** | T | U | U |
|  | M | 53 |  | **1344*** | DT | U | U |
|  | F | 93 |  | 318 | D | 2.3 | VI |
|  | F | 92 |  | 955 | D | 20.5 | III |
|  | F | 56 |  | 4238 | D | 12 |  |
|  | M | 70 |  | 4534 | D | 28 |  |
|  | F | 91 |  | 11488 | D | 16 | II |
|  | F | 79 |  | 13188 | D | 12.5 |  |
|  | F | 90 |  | 13204 | D | 9.5 | II |
|  | F | 103 |  | 60329 | D | 5 | III |
|  | F | 85 |  | 60428 | D | 8.5 | III |
|  | F | 99 |  | 60524 | D | 15 | II |
|  | F | 95 |  | 62043 | D | 20.5 |  |
|  | M | 71 |  | 389 | M | 15 |  |
|  | F | 83 |  | 719 | M | 17 | III |
|  | M | 69 |  | 946 | M | 12 |  |
|  | M | 87 |  | 2039 | M | 6.3 | III |
|  | F | 86 |  | 4546 | M | 22 |  |
|  | M | 91 |  | 60772 | M | 16 | II |
|  | F | 80 |  | 61218 | M | 5.5 |  |
|  | M | 87 |  | 61334 | M | 8 | II |
|  | M | 88 |  | PDC2 | M | U |  |
|  | M | 80 |  | PDC5 | M | U |  |
|  | M | 75 |  | PDC8 | M | U |  |
|  | **Mean** | **79.54** |  |  |  |  |  |

*Table 1. Continued on next page*

*Table 1. Continued*

|  | Sex | Age | Paper code | Sample | Experiments | Post mortem interval | Braak score |
|---|---|---|---|---|---|---|---|
| Non-diseased cebellum, N = 15 | F | 74 | **ND-2** | **1901** | DSB | 2.3 | II |
|  | F | 74 | **ND-8** | **299** | D | 2.8 | II |
|  | F | 84 | **ND-3** | **703** | S | 5.8 | III |
|  | F | 77 |  | 1569 | D | 8 | III |
|  | F | 83 |  | 719 | D | U | U |
|  | F | 53 |  | **1379** | D | 15 | III |
|  | F | 71 |  | **1571** | DT | U | U |
|  | F | 87 | **ND-10** | 1502 | D | 5 | II |
|  | M | 53 |  | **1344** | DT | U | U |
|  | M | 96 | **ND-1** | **1102** | DSB | 3.4 | II |
|  | M | 83 | **ND-5** | **2501** | DB | 1.7 | II |
|  | M | 95 | **ND-4** | **1301** | DS | 3.5 | I |
|  | M | 79 | **ND-9** | **827** | D | U | U |
|  | M | 87 |  | **1471** | D | U | U |
|  | F | 86 |  | 4546 | M | 22 | U |
|  | **Mean** | **78.8** |  |  |  |  |  |
| DS/AD, N = 3 | F | 51 | **DS-1** | M1864 | B | 19 | U |
|  | F | 47 | **DS-2** | M3233 | S | 24 | U |
|  | F | 44 | **DS-3** | 1258 | S | 13 | U |
|  | **Mean** | **47.3** |  |  |  |  |  |
| LYM, N = 21 | F | 40 |  | 5162 | D | N/A | N/A |
|  | F | 40 |  | 3963 | D | N/A | N/A |
|  | F | 63 |  | 4984 | D | N/A | N/A |
|  | F | 60 |  | 4519 | D | N/A | N/A |
|  | M | 55 |  | Lym 1 | D | N/A | N/A |
|  | M | 35 |  | 4651 | D | N/A | N/A |
|  |  |  |  | 29 | D | N/A | N/A |
|  |  |  |  | 187 | D | N/A | N/A |
|  |  |  |  | 4781 | D | N/A | N/A |
|  |  |  |  | 4801 | D | N/A | N/A |
|  |  |  |  | 4903 | D | N/A | N/A |
|  | M | 28 |  | 5259 | D | N/A | N/A |
|  | M | 56 |  | 83 | M |  |  |
|  | F | 52 |  | **1344** | M |  |  |
|  | F | 56 |  | 4603 | M |  |  |
|  | M | 54 |  | 4609 | M |  |  |
|  | M | 58 |  | Lym 2 | M |  |  |
|  | F | 51 |  | Lym 3 | M |  |  |
|  | M | 52 |  | Lym 4 | M |  |  |
|  | **Mean** | **50.0** |  |  | M |  |  |

Samples in bold are paired CBL and CTX.

*Denotes mid frontal gyrus (MFG), D = DNA content analyses, S= Small population qPCR, T = FISH Analysis, B = single cell qPCR on Biomark HD, P=PNA FISH, M = Westra et al. DNA content metadata.

designed in house and optimized to an annealing temperature of 59°C using Primer3 software (University of Massachusetts Medical School) and synthesized by Valuegene (San Diego, CA) (*Table 2*). For microfluidics-based qPCR, TaqMan assays against *APP* exons 3 and 14 (178 kb apart) (designed from NCBI Reference Sequence: NG_007376.1), *SEMA4A* and *PCDH11X* were synthesized by Applied Biosystems (Life Technologies, Carlsbad, CA) or Integrated DNA Technologies (*APP* exon 3) (San Diego, CA) and optimized to an annealing temperature of 60°C. Primers were assessed in silico to determine specificity of the primer set for a single genomic region and for the presence of SNPs in the targeted genomic region which could decrease primer binding and amplification efficiency. The specificity of all qPCR assays was assessed by gel electrophoresis to confirm a single PCR product of the expected length and sequenced to confirm amplification of the expected product.

Standard curves were used to determine the amplification efficiency of all primer sets used for qPCR. For standard qPCR primers, curves were created by serially diluting purified pGEM-T Easy plasmid DNA (Promega, Madison, WI) containing a single copy of the gene of interest (*Pfaffl, 2001*); serial dilutions of genomic DNA were used for TaqMan primers (*D'Haene et al., 2010*). DNA concentrations were converted to gene copy number by calculating the weight (in g/mol) of the DNA used for generating the standard curve. A linear regression of the curve comparing the log of the gene copy number vs the crossing threshold (Ct) of the primer set was determined from primer efficiency (E) = $10^{-1/slope} - 1$ (*Pfaffl, 2001*). Only standard curves with $R^2$ values of greater than 0.99 were used.

## Standard qPCR on small neuronal populations

Genomic DNA from bulk-sorted nuclei (described above) was isolated using the DNeasy Blood and Tissue Kit (Qiagen, Valencia, CA) and quantified using Quant-iT PicoGreen (Life Technologies, Carlsbad, CA); genomic DNA was stored at −20°C before use. Standard qPCR reactions using SYBR Green (Promega, Madison, WI) fluorescence detection (ex: 494 nm; em: 529 nm) were performed in triplicate using 0.5 ng of sample gDNA per reaction. Reactions were run on a Rotor-Gene RG-3000 72-well thermocycler (Qiagen, Valencia, CA) using GoTaq qPCR master mix (Promega, Madison, WI) and the following parameters: denaturation (95°C for 5 min), amplification (95°C for 25 s, 59°C for 30 s and 72°C for 30 s), and quantification through 40 cycles; and a melting curve determination (55–99°C, 30 s on the first step, 5 s for each subsequent step). The crossing threshold (Ct) was determined for each primer set within the linear region of the amplification curve. Down Syndrome nuclei were used as a control.

## Microfluidics-based qPCR on single neuronal nuclei

Single neuronal nuclei from FACS were sorted directly into a 96-well plate containing QuickExtract DNA Extraction Solution (Epicentre, Illumina, San Diego, CA) according to the manufacturer's instructions.

**Table 2.** Primers used for small population and single nuclei qPCR

| Gene | Protein | Locus | Assay type | Primer sequence | Probe | Product length | Efficiency |
|------|---------|-------|-----------|-----------------|-------|---------|-----------|
| *APP*, Exon 14 | Amyloid precursor protein | 21q21.3 | SYBR Green | F-TGCACGTGAAAGCAGTTGAAG, R-AAAGATGGCATGAGAGCATCG | N/A | 214 | 0.973 |
| SEMA4A | Semaphorin 4A | 1q22 | SYBR Green | F- ATGCCCAGGGTCAGATACTAT, R-TTCTCCGAGATCCTCTGTTTC | N/A | 177 | 0.997 |
| CCL18 | Chemokine (C–C motif) ligand 18 | 17q11.2 | SYBR Green | F-TTCCTGACTCTCAAGGAAAGG, R-CTGGCACTTACATGACACCTG | N/A | 209 | 1.006 |
| PCDH11X | Protocadherin 11 X-linked | Xq21.3 | SYBR Green | F-TCTTTTGGTCAGTGTTGTGCG, R-CAACAAGTCGCCTATCAGGAC | N/A | 188 | 0.993 |
| *APP*, Exon 14 | Amyloid precursor protein | 21q21.3 | TaqMan | CGGTCAAAGATGGCATGAGAGCATC*, Assay Hs01255859_cn | FAM-MGB | 91 | 1.040 |
| *APP*, Exon 3 | Amyloid precursor protein | 21q21.3 | TaqMan | F-GCACTTCTGGTCCCAAGCAT, R-CCAGTTCTGGATGGTCACTG | ROX-IB | 140 | 0.992 |
| SEMA4A | Semaphorin 4A | 1q22 | TaqMan | GTTCAAGGGTATGTGAGGTGAGATG*, Assay Hs00329046_cn_VIC | VIC-MGB | 90 | 1.016 |

*Denotes probe sequence provided by Life Technologies.

Multiple independent sorts were completed for each group and each individual. Prior to analysis on the Fluidigm Biomark HD (South San Francisco, CA), single neuron genomic DNA was pre-amplified as per Fluidigm protocols (*Fan and Quake, 2007*; *Dube et al., 2008*; *Qin et al., 2008*; *Jones et al., 2011*; *White et al., 2011*; *Whale et al., 2012*) on a Veriti thermocycler (Life Technologies, Carlsbad, CA) using locus-specific Taq-man primer sets (primer Table above) (20× initial concentration, 18 μM primer, 5 μM probe; combined and diluted to 0.2×) (95°C denaturation for 5 min; 18 amplification cycles of a 95°C denaturation for 15 s, followed by a 60°C annealing and extension step for 4 min; and a final extension step at 72°C for 7 min). Locus-specific pre-amplification was confirmed on a Roche LightCycler (Roche Applied Science, Indianapolis, IN) using one targeted primer set prior to analysis on the Biomark 48.48 Dynamic Array integrated fluidic circuit (IFC) (Fluidigm, South San Francisco, CA). Samples were diluted 1:5 and loaded into the 48.48 Dynamic Array IFC according to the manufacturer's protocol. DNA was loaded in triplicate and assays in sextuplicate for a total of 18 replicates per assay per nucleus. Samples were run across multiple arrays for quality control between runs (*Table 3*) and multiple individuals were run on each array. The thermocycling program was performed on the Biomark: 95°C for 10 min, then 55 cycles of 95°C denaturation, and 60°C annealing and extension. Fluorescent probes used for these assays were 5′-FAM or 5′-VIC with a 3′-minor groove binding (MGB) non-fluorescent quencher, or 5′-ROX with a 3′ Iowa Black quencher. Ct values were determined using Fluidigm's Real-Time PCR Analysis Software. Only nuclei with 10 or more replicates per assay were used for analysis.

## Quantitative PCR analysis

For both qPCR strategies, relative copy number (RCN) for a diploid sample was calculated as $RCN = 2 \times (1 + E)^{-\Delta\Delta Ct}$ where E is primer efficiency ($E = 10^{-1/slope} - 1$) (*Weaver et al., 2010*; *Pfaffl, 2001*; *D'Haene et al., 2010*; *Livak and Schmittgen, 2001*). Paired cerebellar samples were used as a calibrator to determine ΔΔCt values. For single cell RCN determinations, cerebellar ΔCt values for the paired cerebellum were averaged. RCNs were modeled for each copy number 1–6, assuming a system standard deviation of 0.25 and a 95% CI equal to the standard error of the mean multiplied by the critical t value for a two-tailed t-distribution (degrees of freedom = 68) with p = 0.05 (*Weaver et al., 2010*). 95% confidence interval (CI) was also determined for the RCN of each assay for each single nucleus giving $RCN = 2 \times (1 + E)^{-\Delta\Delta Ct} \pm CI$, and the upper and lower bounds were used to call copy numbers for *APP* exons 3 and 14 such that a CI that overlapped with the modeled RCN range was considered as belonging in that copy number bin (See *Table 4* for modeled upper and lower bounds). Copy number bins were 1–6 and >6, as beyond 6 copies the CIs for the given degrees of freedom begin to overlap, allowing for assessments of significant increases in copy number but limiting the ability to distinguish between nuclei with, for example, 8 vs 9 copies. There was high concordance between the *APP* exons 3 and 14 demonstrating minimal amplification bias between the primer sets.

## Statistical analysis

For DNA content FCM, DNA indices (DIs) were determined by taking the ratio of the mean from the diploid (2N) peak from the brain sample to the mean of the lymphocyte control peak, both normalized to the mean of the CEN standard (*Darzynkiewicz and Huang, 2004*; *Darzynkiewicz et al., 2004*; *Westra et al., 2010*). The percent change was calculated from DI values assuming a 2N diploid would have a DI of 1. p values for comparison of mean percent change, skew and coefficients of variation comparisons were determined by one-way ANOVA and Tukey's multiple comparison tests. Linear

**Table 3.** Quality control between 48.48 Dynamic Array runs

|  |  | Cell 1 | Cell 2 | Cell 3 | Cell 4 | Cell 5 | Cell 6 | Cell 7 | Cell 8 |
|---|---|---|---|---|---|---|---|---|---|
| APP 14 | Run 1 | 21.37 | 18.77 | 18.02 | 18.34 | 17.11 | 19.33 | 20.12 | 18.02 |
|  | Run 2 | 20.60 | 18.77 | 18.14 | 18.41 | 17.25 | 19.50 | 20.60 | 18.26 |
| APP3 | Run 1 | 22.79 | 20.35 | 18.53 | 19.11 |  |  |  |  |
|  | Run 2 | 21.97 | 19.84 | 18.65 | 19.28 |  |  |  |  |
| SEMA4A | Run 1 | 25.54 | 24.29 | 22.44 | 24.23 | 22.81 | 25.74 | 24.46 | 24.67 |
|  | Run 2 | 25.26 | 24.53 | 23.92 | 24.39 | 23.03 | 25.42 | 24.11 | 24.30 |

**Table 4.** Confidence Intervals for (CI) calling Copy Number (CN)

| CN | CI | RCN, value |
|----|-------|------------|
| 1 | Lower | 0.92156 |
|   | Upper | 1.08512 |
| 2 | Lower | 1.84312 |
|   | Upper | 2.17023 |
| 3 | Lower | 2.76468 |
|   | Upper | 3.25535 |
| 4 | Lower | 3.68624 |
|   | Upper | 4.34047 |
| 5 | Lower | 4.60780 |
|   | Upper | 5.42559 |
| 6 | Lower | 5.52936 |
|   | Upper | 6.51070 |

regression analysis was used to determine age-percent change correlation. For comparison of average percent change from NeuN-positive vs negative cortical nuclei, or NeuN-positive cortical and cerebellar nuclei, p values were determined by unpaired, two-tailed t-test. For standard qPCR analyses, differences in ΔΔCt ± SEM of *APP* in the cortex vs cerebellum were assessed in each individual using an unpaired, two-tailed t-test. For single cell qPCR, p values were determined by one-way ANOVA and Tukey's multiple comparison tests.

## Fluorescent in situ hybridization

Isolated nuclei were stained with DAPI and hybridized using dual color FISH as described previously (*Rehen et al., 2001*; *Kaushal et al., 2003*). FISH paints against the whole q arm of chromosome 21 and a point probe against a region on the q arm of 21 (21q22.13-q22.2) (Vysis. Downer's Grove, IL) were used. The mounted slides were examined on a Zeiss Axioskop microscope and Axiocam CCD camera (Carl Zeiss, Thornwood, NY). Approximately, 500 nuclei were blindly counted for each brain, by two independent observers, on 14 samples (5 non-diseased, 9 diseased). Total AD cortical nuclei were examined using a highly liberal protocol for calling aneusomies whereby borderline FISH profiles suggestive of aneusomy were always included in quantitative assessments. All analyses were conducted blind to the identity of samples by interrogating purified nuclei. The ability of this technique to detect aneuploidy was validated using interphase nuclei from a human trisomy 21 brain (*Svendsen et al., 1998*) revealing three nuclear signals (*Figure 5D,E*).

## PNA probe design and dot blot confirmation

Peptide nucleic acid probes were custom designed in coordination with PNA Bio, Inc. Nucleic acid sequences were identified and analyzed in silico to ensure binding to only one specific genomic region on *APP*. Nine unique probes (4 on exon 3 and 5 on exon 14) were designed and conjugated to a single fluorophore of Alexa-488 (*Table 5*). Specificity of probes was confirmed using dot blot DNA detection paired with immunoblot via antibodies against the Alexa-488 fluorophore (AF-488) (*Figure 5—figure supplement 1A*). Increasing DNA concentration (1.8, 3.6 and 5.4 μg) of plasmids containing one copy of each exon (all nine PNA binding sites) was used to verify linear increases in AF-488 signal (*Figure 7—figure supplement 1B*). For the APP copy number curve, DNA concentration remained consistent but plasmids containing 0, 3, 6, and 9 copies of all APP PNA binding sites were used. DNA was denatured in 0.1 M NaCl at 50°C and dot blotted onto a positively charged nylon membrane in distilled deionized water (DDW) and washed in 6× SSC. DNA was cross linked to the membrane, and PNA probes were hybridized to the membrane using conditions consistent with probe hybridization to nuclei on slides. Briefly, probes were prepared in 20 mM Tris, 60% formamide, and 0.1 μg/ml salmon sperm. Probes and membrane were heated 85°C for 5 min, and probes were added to the membrane

**Table 5.** Peptide nucleic acid (PNA) probe sequences

| Gene | Protein | Locus | Assay | Sequence |
|------|---------|-------|-------|----------|
| APP Exon 3 | Amyloid precursor protein | 21q21.3 | PNA FISH | A488-GATGGGTCTTGCACTG, A488-CCCCGCTTGCACCAGTT, A488-GGTTGGCTTCTACCACA, A488-CAGTTCAGGGTAGAC |
| APP Exon 14 | Amyloid precursor protein | 21q21.3 | PNA FISH | A488-CTCCATTCACGG, A488-GTGGTTTTCGTTTCGGT, A488-ACTGATCCTTGGTTCAC, A488-ACTGATCCTTGGTTCAC, A488-ACGTCATCTGAATAGTT |
| Telomere | N/A | Telomeres | PNA FISH | TelC-Cy3 (F1002, PNA BIO) |

and incubated at 85°C for 10 min, followed by room temperature overnight. Membrane was then washed twice at 60°C in 2× SSC + 0.1% Tween-20, and then in successive washes of 2× SSC, 1× SSC, and DDW. Probes were visualized on the membrane using a Typhoon fluorescence scanner from General Electric (Fairfield, CT). For quantification, color data were removed, image was inverted, brightness/contrast was adjusted evenly across the entire image, and average pixel intensity was acquired for a region of interest with a standardized area across sample comparisons.

### Peptide nucleic acid fluorescent in situ hybridization

Probes were hybridized according to the manufacturer's instructions (*Lansdorp et al., 1996*). Neuronal nuclei from non-diseased and AD brains were sorted for NeuN positivity and dried onto slides. Slides were washed in PBS, fixed with 4% PFA, and treated with RNase (Qiagen, Valencia, CA) for 20 min at 37°C. Slides were then digested with 200 µg/ml proteinase K (Roche Applied Science, Indianapolis, IN) for 5 min at 37°C. Slides were dehydrated in ethanol series and denatured at 85°C for 5 min. PNA probes in hybridization buffer (20 mM Tris pH 7.4, 60% formamide, 0.1 µg/ml salmon sperm DNA) were then added for 10 min at 85°C. Slides were then removed and placed at room temperature for 2 hr. Slides were then washed twice in 2× SSC + 0.1% tween at 60°C for 30 min each and mounted using progold mounting media. Z-stacks were acquired using a Nikon N-SIM (structured illumination microscopy) super resolution microscope with an Andor iXon3 back-illuminated high sensitivity EMCCD camera with single photon detection capability. Projections were rendered using 3D-SIM Elements software from Nikon. Counts were averaged and analyzed by unpaired, two-tailed t-test

## Acknowledgements

We thank Drs Don Cleveland, Edward Koo, and Kun Zhang for helpful comments and suggestions; Drs Olga Pletnikova and Juan Troncoso at the Johns Hopkins School of Medicine Alzheimer's Disease Research Center for providing human brains; the Flow Cytometry Core at TSRI for FACS expertise; Hope Mirendil for help with qPCR and manuscript comments; Mr Mathew McCreight for FISH quantitation; Uma Dandakar and Hemani Wijesuriya of the UCLA GenoSeq Core; Kathy Spencer, Michael Thieleking, and Ariana Lorenzana of The Nikon Center of Excellence at TSRI; Caroline Dando and Susan Robelli of Fluidigm for advice on Fluidgm usage; Andrew Richards for advise on WGA experiments; and Danielle Jones for manuscript editing. This work was partially funded by the NIH (MH076145 to JC), (T32 DA07315 to DMB and GEK), (T32 AG 000216 to GEK), The Shaffer Family Foundation and private sources (to BS, RRR and JC). The UCI-ADRC is funded by NIH/NIA Grant P50 AG16573.

## Additional information

### Funding

| Funder | Grant reference number | Author |
| --- | --- | --- |
| National Institutes of Health | MH076145 | Jerold Chun |
| Shaffer Family Foundation | | Benjamin Siddoway, Richard R Rivera, Jerold Chun |
| National Institutes of Health | T32 DA07315 | Diane M Bushman, Gwendolyn E Kaeser |
| National Institutes of Health | T32 AG 000216 | Gwendolyn E Kaeser |

The funders had no role in study design, data collection and interpretation, or the decision to submit the work for publication.

### Author contributions

DMB, GEK, BS, JWW, RRR, SKR, YCY, JC, Conception and design, Acquisition of data, Analysis and interpretation of data, Drafting or revising the article

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
