## [Decision Letter]

Thank you for sending your work entitled “Genomic mosaicism with increased *APP* copy number in single neurons from sporadic Alzheimer's disease brains” for consideration *at eLife*. Your article has been favorably evaluated by a Senior editor, a Reviewing editor, and 3 reviewers.

The following individuals responsible for the peer review of your submission have agreed to reveal their identity: Jeremy Nathans (Reviewing editor), Dimitri Avramopoulos, and Phil Wong (peer reviewers). A further reviewer remains anonymous.

The Reviewing editor and the other reviewers discussed their comments before we reached this decision, and the Reviewing editor has assembled the following comments to help you prepare a revised submission. I am also including the three reviews in their original form at the end of this letter, as there are many specific and useful suggestions in them that will not be repeated in the summary here.

We would like to request that you resubmit a revised manuscript that addresses the specific issues raised in the reviews below. All three reviewers raised substantive concerns, but the ones that we see as central relate to: (9) the comparison of AD vs. normal brains (review #2), and (2) the question of whether the changes in genomic DNA associated with nuclei from AD brains is a cause or consequence of the disease process (review #1). We recognize that assessing levels of *APP* expression, while logically related to your hypothesis, is not easily done on a single cell level in postmortem brain samples. Even if one assumes that the data is technically sound, there is a major concern that DNA content changes may not be causally related to AD.

A critical discussion of potential artifacts in the data and of the biological interpretation of the data is essential. We note that in the current publishing environment very few authors write truly critical analyses of their experiments. It was not always that way, and *eLife* is trying to restore some balance here. We welcome a critical Discussion section, which we believe enhance the reader's ability to judge science. Here is a nice example that illustrates the preceding point: it is from the paper describing the discovery of ubiquitin conjugation as the mark for protein degradation (Hershko A. et al., Proposed role of ATP in protein breakdown: conjugation of protein with multiple chains of the polypeptide of ATP-dependent proteolysis, PNAS, 1980, April 77 (48): 1783-6). After summarizing the evidence that conjugation of APF–1 (=ubiquitin) marks proteins for degradation, the authors write: “Evidence that APF–1-proteins are intermediates in the breakdown of denatured protein as proposed in Figure 6 is indirect and inconclusive at this time.”

*Reviewer #1*:

“The validated pathogenicity of *APP* in familial AD suggested that mosaic alterations in *APP* copy number within single neurons may be a key mechanism for producing sporadic AD. Through the use of five independent experimental approaches, we report increased somatic genomic variation within individual sporadic AD neurons involving mosaic increases in both DNA content and *APP* copy number.”

The premise of the manuscript by Bushman et al., as stated above, is to suggest that a mechanistic basis for sporadic Alzheimer's disease may be the consequence of increased levels of expression of *APP* in neurons of the cerebral cortex due to mosaic increases in *APP* copy number. The validation of this premise would provide an impetus to further mechanistic studies in AD as well as suggest strategies for therapeutic intervention. Therefore, it is important that the premise be validated in a rigorous manner. There are fundamental challenges to this effort, however. Critical to validation of this premise is the establishment of the presence of the additional *APP* copies in neurons well before the demise of the individual contributing the nuclei of their cortical neurons to the study. An interpretation of the data presented which must be considered is the possibility that the additional DNA present in the nuclei analyzed occurred in the terminal phase of the life of these nuclei shortly before or at the time of death and thus have no direct connection to the pathology experienced by these neurons. Associated with this issue is an understanding of the precise form the observed additional DNA has taken in the analyzed nuclei which would address two questions: first, are complete additional copies of *APP* genomic DNA in fact present in a contiguous fashion and, second, what mechanistic basis might be associated with the occurrence of these additional copies of *APP* DNA. A third approach which might be taken to validate the premise would be to carry out studies directed at evaluating *APP* expression levels in individual neurons to address the presumed pathological consequence of amplification of the *APP* gene in cortical neurons in AD.

The study could have addressed some of these issues but is currently limited in its present form. I will address each of the sets of data presented individually.

Figure 2: the data shows that a significant number of nuclei contain additional DNA sequences, approximately 7% of additional DNA in a subset of cortical nuclei.

While this reviewer finds the finding that additional DNA is present in a subset of nuclei derived from AD cortex to be persuasive, nothing in the data convinces me that the DNA sequences present in these nuclei were all present in these nuclei at the last point in time that the nuclei were present inside intact neurons. However, this reviewer suggests that the authors carry out whole genome sequencing on a population of nuclei sorted for increased DNA content. What would be the value of such sequencing:

1) The premise is that *APP* sequences are preferentially increased in copy number. This sequencing should confirm that *APP* gene copy number is increased in these nuclei. The specificity of this increase, and some potential mechanistic insight into the process which produces it, should emerge from an evaluation of what other sequences are increased in copy number besides *APP*.

2) If intact and functional copies of the *APP* gene are present in the nuclei of the AD cortical neurons examined, then a signature of the mechanistic basis by which the presumed amplification had occurred should be present. This might include the observation of junction sequences between the *APP* gene flanking regions and some other regions in the genome, indications of *APP* gene circularization or other mechanisms that would permit the stable presence of the amplified copies of the *APP* gene to persist in the nuclei of cortical neurons for a substantial period of time.

Figure 3: Neu-N positive staining is a reasonable criteria for ascertaining that the nuclei under analysis were in fact once present in neuronal cells. It does not, however, establish that at the point in time that additional *APP* gene containing DNA appeared in this nucleus, that the nucleus was present inside an intact cell membrane.

Figure 4: small pool qPCR assays are consistent with the view that there are increased relative amounts of *APP* gene DNA in the samples derived from AD cortical neurons but give no further insight into how they got there and when. The FISH studies are clear, eliminating the possibility of acquisition of additional complete copies of chromosome 21 as the source of additional *APP* gene copies and establishing the competence of the investigators to carry out two color FISH with genomic DNA probes of significant length.

Figure 5: these data are consistent with the small pool data for Figure 4. A useful extension of this data would have been to continue to carry out PCR for DNA sequences beyond exon 3 and exon 14 into the upstream and downstream regions of the *APP* gene. The point is for each case analyzed the size of the DNA segment present in increased copy number is less than an entire chromosome 21 but at least equal in size to the full APP gene transcriptional unit. The DNA samples prepared for this study can easily be quantitated for additional sequences in the *APP* gene region of chromosome 21 to establish where the beginning and end of the genomic sequence of increased copy number is to be found. Are there discrete ends to the increased copy number sequence encompassing exons 3 through 14 of *APP*? If so what might that information tell you about the mechanistic basis for increased copy number?

Figure 6: the investigators turn to PNA based in situ hybridization to permit visualization of the *APP* genomic sequences exhibiting increased copy number. The authors justify this switch in technology from the techniques used in Figure 4 based on the following statement: “However, conventional point-probe FISH analysis, while useful for aneusomy assessments, cannot quantitatively evaluate copy number changes in single gene loci,…” (Results section).

As I noted above in relation to Figure 4, this statement is not correct. In fact the author's use a 330 kb FISH probe (the approximate size of the *APP* locus) and successfully and accurately assess its copy number in the studies shown in that figure. The use of genomic DNA probes of lengths of 100 kbp or less is and has been routine for in situ hybridization for detecting chromosome rearrangement and breakpoints and copy number for decades. The switch to PNA probes of very limited length adds a layer of concern about the studies shown in Figure 6. The authors have the capability to do this study as it should be done. A great deal of information on the structure of the extra copies of the *APP* gene can be gained by doing a proper study with conventional genomic DNA probes. They should do this and compare the results of a systematic analysis using probes which extend through and beyond the *APP* gene to address the same questions which can be addressed by extending the PCR based studies to the edges of the *APP* gene and beyond.

In summary, the conceptual basis of these studies is of interest. However, by repeatedly reciting that they have proved the functional significance of their observations by five different methods instead of candidly addressing the limitations of their studies and the possibility that their data may have no bearing whatever on the mechanistic basis of sporadic Alzheimer's disease, I find the manuscript unsatisfactory for publication in its present form.

*Reviewer #2*:

This is a very interesting paper showing evidence in support of brain specific somatic mosaicism for duplications and amplifications of genetic sequences and in particular for the Alzheimer's disease (AD) gene *APP*. While *APP* is a known early onset AD gene through both non-synosynous mutations and copy number variation, this work introduces the concept that mosaicism for amplification of *APP* in the cortex could underlie late onset sporadic AD. If true, this represents a very significant progress in our understanding of sporadic AD and given the disease prevalence it can also lead to a major public health impact.

The paper is very well written and easy to understand, the experiments are appropriate and well described, and multiple approaches are used to confirm each result.

Through their work the authors provide convincing evidence that increase in DNA content, which appears to be due to genomic amplifications in many loci including the *APP* gene locus, is a common phenomenon in the cortex, less common in the cerebellum and uncommon in lymphocytes. This is very interesting and indeed provides a possible mechanism for the pathology of sporadic AD. However, no significant evidence is shown to support that this phenomenon is more prevalent in the AD cortex than controls. The presented data do show a somewhat more pronounced effect in AD but the similar, albeit weaker, finding in controls is downplayed, and the data is mostly placed in the supplementary material. Statistical comparisons between case and control brains that could convince the reader that this is indeed more pronounced in AD brains are not provided and would likely not be significant due to the sample size. While it is likely that there is a true difference, the authors should be more cautious about making statements of “mosaicism in AD brains”, which implies that this is not the case in controls. They should also provide more discussion on the fact that this is also observed in cases. While this phenomenon provides a substrate that might be necessary or just facilitate the development of plaques in AD, based on the authors own data it is likely not sufficient. These points should be made in the Discussion.

The manuscript would also benefit from more discussion on the significant differences between lymphocytes, cerebellum and cortex. It seems perhaps counter intuitive that cells that undergo many divisions throughout life maintain their DNA content much better than cells that do not, and some discussion on this would be useful in better understanding the results and their potential impact.

*Reviewer #3*:

By exploiting the observation that individual neurons in brains display somatic genomic mosaicism, Bushman et al. showed such phenomenon of increases in DNA content and specific elevation of copy number of the *APP* gene occurred in single neurons from affected regions of cases of sporadic Alzheimer's disease (AD). These novel findings would have important implications for *APP* gene dosage as a critical determinant of sporadic AD, for which its etiology remain largely elusive.

While the authors provide a set of compelling evidence using state of the art approaches to support their conclusion, one assumption is that the increase of up to 12 copies of *APP* gene in neurons from AD are all functional and active genes. As previous work has documented that the coding region of human APP gene spans ∼400 kb of genomic DNA (Rooke et al., Mamm. Genome, 4:662-9, 1993), it is possible that the increase in *APP* gene copy number in neurons from AD cases do not represent active full length *APP* genes. Direct evidence to distinguish whether active full length copies of *APP* gene from inactive non-transcribed pseudogenes account for the increase in gene copy number in neurons will strengthen this manuscript. For example, is there any evidence that the *APP* transcript (as judged by in situ rtPCR using RNAscope technology) is increased in these neurons with elevation in copy number of *APP* gene? Positive data will strongly support the notion that the increase in amyloid burden can be attributable to the increase in number of active *APP* gene in neurons within affected regions of sporadic AD patients.

---

## [Author Response]

*We would like to request that you resubmit a revised manuscript that addresses the specific issues raised in the reviews below. All three reviewers raised substantive concerns, but the ones that we see as central relate to: (*[9]*) the comparison of AD vs. normal brains (review #2), and (*[2]*) the question of whether the changes in genomic DNA associated with nuclei from AD brains is a cause or consequence of the disease process (review #1). We recognize that assessing levels of* APP *expression, while logically related to your hypothesis, is not easily done on a single cell level in postmortem brain samples. Even if one assumes that the data is technically sound, there is a major concern that DNA content changes may not be causally related to AD.*

The first major concern was the comparison of AD vs. normal brains (i.e., case vs. control): this has now been addressed by experimentally increasing the N for non-diseased brains (11 new brains), controlled by repeats of AD samples from the original cohort, to improve statistical power of our data. Increased significance was achieved to differentiate diseased vs. non-diseased cortical samples (P=0.0034). In addition, we also pursued meta-data analysis of previously analyzed, non-diseased cortical samples to still further increase the “N” of non-diseased samples, towards better assessing statistical separation between non-diseased and diseased samples. This approach was made possible by use of an independent, previously assessed age-matched cohort of non-diseased brains that had been studied using precisely the same protocols and flow cytometry equipment (from [74]) as that used in the current manuscript, whereby raw data from flow cytometry were re-analyzed identically to: 1) the previous data of the first submission and 2) new data processed in response to the review. These meta-data points are distinguished by distinct icons in new Figure 3. Notably, new samples without meta-data demonstrate statistical significance (P=0.0034), and with inclusion of the meta-data, even more statistical significance (P=0.0001). These data demonstrate statistical separation between AD vs. non-diseased cortex. Further details are provided in the point-by-point critique.

The second concern related to DNA content and causality is clearly of great import, yet also extremely difficult to address experimentally in human post-mortem studies. Further research will be necessary in order to answer “chicken or egg” questions, as well as those related to function and pathogenicity: an historical case in point is ApoE that was identified as an AD risk factor over 20 years ago yet for which function and mechanistic pathogenicity is still not completely understood. Similarly, even in the case of familial forms of AD involving defined heritable gene mutations present throughout life, decades are still required for the clear manifestation of disease, thus underscoring our incomplete understanding of even well studied pathogenic AD genes like *APP*.

To acknowledge this deficiency in understanding that extends from familial disease through our current report, we have now both softened the writing throughout the paper and have added a discussion of our study’s limitations.

*A critical discussion of potential artifacts in the data and of the biological interpretation of the data is essential. We note that in the current publishing environment very few authors write truly critical analyses of their experiments. It was not always that way, and* eLife *is trying to restore some balance here. We welcome a critical Discussion section, which we believe enhance the reader's ability to judge science. Here is a nice example that illustrates the preceding point: it is from the paper describing the discovery of ubiquitin conjugation as the mark for protein degradation (Hershko A. et al., Proposed role of ATP in protein breakdown: conjugation of protein with multiple chains of the polypeptide of ATP-dependent proteolysis, PNAS, 1980, April 77 (4): 1783-6). After summarizing the evidence that conjugation of APF-1 (=ubiquitin) marks proteins for degradation, the authors write: “Evidence that APF-1-proteins are intermediates in the breakdown of denatured protein as proposed in*
Figure 6
*is indirect and inconclusive at this time.”*

We welcome the opportunity to provide a critical Discussion section and have now added this into the text.

Reviewer #1:

*“The validated pathogenicity of* APP *in familial AD suggested that mosaic alterations in* APP *copy number within single neurons may be a key mechanism for producing sporadic AD. Through the use of five independent experimental approaches, we report increased somatic genomic variation within individual sporadic AD neurons involving mosaic increases in both DNA content and* APP *copy number.”*

*The premise of the manuscript by Bushman et al., as stated above, is to suggest that a mechanistic basis for sporadic Alzheimer's disease may be the consequence of increased levels of expression of* APP *in neurons of the cerebral cortex due to mosaic increases in* APP *copy number. The validation of this premise would provide an impetus to further mechanistic studies in AD as well as suggest strategies for therapeutic intervention. Therefore, it is important that the premise be validated in a rigorous manner. There are fundamental challenges to this effort, however. Critical to validation of this premise is the establishment of the presence of the additional* APP *copies in neurons well before the demise of the individual contributing the nuclei of their cortical neurons to the study. An interpretation of the data presented which must be considered is the possibility that the additional DNA present in the nuclei analyzed occurred in the terminal phase of the life of these nuclei shortly before or at the time of death and thus have no direct connection to the pathology experienced by these neurons.*

We thank the reviewer for his/her comments. We believe that it is most unlikely that the observed changes in both increased DNA content and increased *APP* copy number are artifacts of death, unrelated to pathology, for multiple reasons:

1) If the changes were non-specifically occurring “before or at the time of death,” DNA degradation should be observed (e.g*.,* nucleosomal “ladders” seen with apoptosis, or genomic DNA degradation seen with sub-optimal post-mortem intervals). The opposite was observed.

2) Similarly, by the critique’s hypothesis, the same degradative phenomenon would have been expected to be observed in non-diseased brains. By contrast, there was statistically significant separation between non-diseased and diseased samples (prior version and new Figure 3).

3) Critically, postulated changes produced by death would be expected to occur uniformly, in all parts of the brain from the same individual. By contrast, the cerebellum always showed less of an effect compared to the cortex in the same individual.

DNA content variation and *APP* copy number increases were also observed, albeit to a lesser extent, in non-diseased samples: our samples were from aged individuals (average age, 82.5 years) supporting the possibility that genomic changes in non-diseased cells might represent a pre-morbid state in some individuals, although this is impossible to prove using post-mortem analyses. However, our data are consistent with the documented presence of senile plaques in normal aged brains (since we observe some *APP* CNVs there), as well as the universal presence of senile plaques in sporadic AD despite no obvious germ line alterations to *APP* or increases in constitutive *APP* copy number.

*Associated with this issue is an understanding of the precise form the observed additional DNA has taken in the analyzed nuclei which would address two questions: first, are complete additional copies of* APP *genomic DNA in fact present in a contiguous fashion and, second, what mechanistic basis might be associated with the occurrence of these additional copies of* APP *DNA.*

We have begun to investigate this more thoroughly, being of absolute interest. However, a comprehensive assessment of all DNA differences, including a demonstration of their pathogenicity, along with the mechanisms that cause genomic mosaicism in neurons (e.g*.,* how and when) is a monumental undertaking, especially in trying to establish statistically significant linkage to any identified sequence differences. We currently have two key hints regarding the “form” of additional DNA copies. First, our single cell qPCR data show a high, but notably non-identical, concordance rate between the two assessed *APP* exons that are separated by 170 Kb, suggesting that the full coding region of *APP* could be amplified rather than only individual exons, although the latter is a non-mutually exclusive possibility as well. The data are thus consistent with increases in the *APP* coding region, along with partial amplification involving one or the other of the two exons. Critically, our PNA-FISH data identify quantitative, often single signals within a nucleus, supporting *APP* amplification within a physically constrained locus.

*A third approach which might be taken to validate the premise would be to carry out studies directed at evaluating* APP *expression levels in individual neurons to address the presumed pathological consequence of amplification of the* APP *gene in cortical neurons in AD.*

Please see response to Review #3.

*The study could have addressed some of these issues but is currently limited in its present form. I will address each of the sets of data presented individually*.

Figure 2*: the data shows that a significant number of nuclei contain additional DNA sequences, approximately 7% of additional DNA in a subset of cortical nuclei*.

*While this reviewer finds the finding that additional DNA is present in a subset of nuclei derived from AD cortex to be persuasive, nothing in the data convinces me that the DNA sequences present in these nuclei were all present in these nuclei at the last point in time that the nuclei were present inside intact neurons*.

Our data argue for the DNA being present before and unrelated to the technique of nuclear isolation. We would like to clarify the methods used. Whole brain tissue (with intact neurons and glia) was first triturated on ice in an NP40-containing buffer used to isolate all brain nuclei. New Figures 2 and 3 (old Figure 2) describe the DNA content shift of whole brain nuclei stained for propidium iodide, while (now) Figure 4 demonstrates that upon immunolabeling of these nuclei for a highly validated neuronal marker, NeuN, the right shifted nuclei are largely neurons. We do not believe that DNA would accumulate within nuclei after they have been isolated for multiple reasons:

1) As noted above, degradative processes degrade DNA rather than produce more synthesis.

2) Isolated nuclei were fixed in seconds following trituration, with 2% paraformaldehyde (and other fixatives have been tried), producing the same results as unfixed samples, and do not support active enzymatic synthesis associated with nuclei isolation methods.

3) Similarly, neurons (NeuN-positive nuclei) have more DNA compared to non-neuronal cells (NeuN-negative) from the same cortical sample, arguing against a generic artifact affecting all isolated nuclei.

4) Hypothesized DNA synthesis after isolation is not consistent with its relative absence in cerebellar neurons from the same individual, as compared to the prefrontal cortical neurons, arguing against a non-specific phenomenon that increases DNA content with nuclear isolation.

5) Published work (76) demonstrated increased DNA content by fluorescence quantitation of centromere sequences (CENP-B box) in single nuclei, consistent with genomic origins of the DNA (rather than other sources like mitochondria or bacterial contamination): these studies were also pursued on fixed nuclei.

However, this reviewer suggests that the authors carry out whole genome sequencing on a population of nuclei sorted for increased DNA content. What would be the value of such sequencing:

*1) The premise is that* APP *sequences are preferentially increased in copy number. This sequencing should confirm that* APP *gene copy number is increased in these nuclei. The specificity of this increase, and some potential mechanistic insight into the process which produces it, should emerge from an evaluation of what other sequences are increased in copy number besides* APP*.*

*2) If intact and functional copies of the* APP *gene are present in the nuclei of the AD cortical neurons examined, then a signature of the mechanistic basis by which the presumed amplification had occurred should be present. This might include the observation of junction sequences between the* APP *gene flanking regions and some other regions in the genome, indications of* APP *gene circularization or other mechanisms that would permit the stable presence of the amplified copies of the* APP *gene to persist in the nuclei of cortical neurons for a substantial period of time.*

We absolutely agree with the reviewer that sequencing of an appropriate template—single cell or cohort population (as suggested—could provide important information on copy number and junctional sequences, and we are endeavoring to identify the appropriate template, which turns out to be difficult because of both biological challenges associated with mosaicism, and technical challenges of single-cell sequencing. Some details underscoring the challenges:

1) The inherent nature of mosaicism is that a population consists of a complex mixture of templates consisting predominantly of germline sequences intermixed with myriad mutant templates (based upon data in our manuscript showing diverse *APP* copies by qPCR and PNA FISH indicative of diverse genomic structures). Bulk sequencing of such a population will report a normal genome, since the rare mutants would be swamped by the germline sequence of unaffected cells as well as the normal allele in a mutant cell. Also, sequencing of mosaic templates does not lend itself to routine CNV calling, as current algorithms (e.g., those used by Illumina) are not capable of making reliable calls on complex, mosaic templates.

2) The reviewer suggests sequencing sorted nuclei to identify full-length *APP* copy number changes or perhaps flanking sequences towards assessing mechanisms of CNV generation. At this time, it is not possible to obtain enough nuclei known to contain specific genomic changes, even assuming that a stereotyped change exists. Moreover, such relatively rare nuclei would require amplification to achieve sufficient sequencing depths to identify mechanistic signatures like flanking sequence changes. Amplification of the template greatly increases noise, and decreases the ability to call copy numbers with certainty.

3) Some groups, including our own, have begun to work out the steps necessary for identifying mosaic changes in single neurons using sequencing [45]; [28]; [11]). However, this technique also has many challenges including: amplification error, extremely low depth (at best a ∼0.025X coverage), a resolution of 2-10 Mb (the *APP* locus is ∼0.3 Mb), as well as inconsistencies in the methods used that limit comparisons between studies, such as different cell and/or nuclei isolation protocols, multiple amplification techniques, and multiple informatics analysis pipelines. Such challenges will likely be overcome in the future, but cannot be met currently with the available technology.

Biological mosaicism and technological limitations on sequencing therefore prevent rigorous copy-number analyses as well as reliable junctional sequence data. However, we believe our current study provides data that represent a starting point for now understanding the sequence changes underlying DNA content variability in a neurodegenerative disease affecting a specific, highly validated AD gene, *APP,* and we expect future studies to sort-out many of these problems.

Figure 3*: Neu-N positive staining is a reasonable criteria for ascertaining that the nuclei under analysis were in fact once present in neuronal cells. It does not, however, establish that at the point in time that additional APP gene containing DNA appeared in this nucleus, that the nucleus was present inside an intact cell membrane*.

We interpret this part of the critique as proposing a process whereby additional *APP* copies would be produced in isolated nuclei, once outside of the cell. The overall issue was discussed in response to the Figure 2 data critique, whereby degradative processes should degrade DNA and be non-specific to thus affect all samples uniformly, which was not the case. This same argument extends to *APP* copies, and once again, we observe comparatively selective increases in the AD cortex that are statistically distinct from the cerebellar cells of the same individual, as well as in both cortex and cerebellum of the non-diseased brain. We therefore believe that increases in *APP* copies are not artifacts of nuclear isolation.

Figure 4*: small pool qPCR assays are consistent with the view that there are increased relative amounts of* APP *gene DNA in the samples derived from AD cortical neurons but give no further insight into how they got there and when. The FISH studies are clear, eliminating the possibility of acquisition of additional complete copies of chromosome 21 as the source of additional* APP *gene copies and establishing the competence of the investigators to carry out two color FISH with genomic DNA probes of significant length.*

Establishing *when* and *how* additional DNA including copies of *APP* appeared in the nucleus is not feasible using post-mortem human tissue and is beyond the scope of this study, although this is of absolute future interest.

Figure 5*: these data are consistent with the small pool data for*
Figure 4*. A useful extension of this data would have been to continue to carry out PCR for DNA sequences beyond exon 3 and exon 14 into the upstream and downstream regions of the* APP *gene. The point is for each case analyzed the size of the DNA segment present in increased copy number is less than an entire chromosome 21 but at least equal in size to the full* APP *gene transcriptional unit. The DNA samples prepared for this study can easily be quantitated for additional sequences in the* APP *gene region of chromosome 21 to establish where the beginning and end of the genomic sequence of increased copy number is to be found. Are there discrete ends to the increased copy number sequence encompassing exons 3 through 14 of* APP*? If so what might that information tell you about the mechanistic basis for increased copy number?*

We agree with the reviewer that this could have been a useful experiment, and such information could provide useful mechanistic insight. Unfortunately, the entire DNA template is consumed in the single-cell Biomark HD qPCR reaction, preventing us from conducting further analyses of the cells already interrogated. We will continue analyses as we improve cell selection strategies in future work, but cannot address this particular issue in the current manuscript.

Figure 6*: the investigators turn to PNA based* in situ *hybridization to permit visualization of the* APP *genomic sequences exhibiting increased copy number. The authors justify this switch in technology from the techniques used in*
Figure 4
*based on the following statement: “However, conventional point-probe FISH analysis, while useful for aneusomy assessments, cannot quantitatively evaluate copy number changes in single gene loci…” (Results section)*.

*As I noted above in relation to*
Figure 4*, this statement is not correct. In fact the author's use a 330 kb FISH probe (the approximate size of the* APP *locus) and successfully and accurately assess its copy number in the studies shown in that figure. The use of genomic DNA probes of lengths of 100 kbp or less is and has been routine for* in situ *hybridization for detecting chromosome rearrangement and breakpoints and copy number for decades. The switch to PNA probes of very limited length adds a layer of concern about the studies shown in*
Figure 6*. The authors have the capability to do this study as it should be done. A great deal of information on the structure of the extra copies of the* APP *gene can be gained by doing a proper study with conventional genomic DNA probes. They should do this and compare the results of a systematic analysis using probes which extend through and beyond the* APP *gene to address the same questions which can be addressed by extending the PCR based studies to the edges of the* APP *gene and beyond.*

We believe that there is some confusion over what we meant by quantifying copy number, which explains our development and use of PNA-FISH. Several points should be considered:

1) Conventional point-probe FISH, as noted in the review and as we used to interrogate trisomy 21 (old Figure 4, now Figure 5), cannot evaluate copy number changes if those changes are occurring as contiguous copies in close proximity to one another. This reflects multiple factors including: the probe itself, that consists of many fluorescently labeled fragments that can have varied melting temperatures (e.g., GC-rich regions); unknown efficiency of fluorochrome conjugation; unknown relationship of a fluorochrome to a given fragment; resonance energy quenching of fluorescence by neighboring fluors; unknown intact state of the targeted locus; etc. One can count fluorescent “dots” on chromosomes, which is routinely done for FISH (and which we used to interrogate chromosome 21), but this approach cannot quantitatively report (in a reliable way) template copies—e.g., multiple copies of a target—within a single dot. Moreover, routine FISH typically uses microscopy that is non-quantitative vs. the SIM approach that we employed.

2) By contrast, PNA FISH provides single-nucleotide sequence specificity and quantitative capabilities. Each PNA probe was designed to limit resonance quenching, and each probe contains only one fluor (Alexa 488). This provided the capability to quantify the fluorescence within a single dot when combined with SIM. PNA-FISH is a validated technology for quantitating telomeric repeats, and moreover, our controls demonstrate specificity for the targeted APP exons as well as the expected quantitative, linear behavior in detecting DNA copies (now Figure 7—figure supplement 1). In addition, PNA-FISH has been used for other genomic loci including CENP-B (Chen et al., 1999, and [76]), alpha satellite domains (Chen et al., 2000), and CAG repeats (Brind’Amour et al., 2012). It is this quantitative information to which we refer: it is fundamentally different from the standard of counting dots as evidence of copies for a single allele, and cannot be approached using standard FISH probes.

3) As importantly, unlike standard FISH, our PNA probes could be and were designed against the same exons as those interrogated by single-cell qPCR, and notably, both techniques produced similar-appearing distributions of differential copy numbers.

4) In view of a) the extensive literature utilizing quantitative PNA techniques to identify genomic loci, b) the specificity and quantitative controls provided in Figure 7—figure supplement 1 (old Figure 6—figure supplement 1), c) the concordance between our PNA-FISH data and qPCR data, and d) the internal consistency of our data that rules out trisomy 21 as increased in AD (and is therefore consistent with local amplification), we believe the employed assay is appropriate and not interchangeable or replaceable by standard genomic FISH.

5) Finally, it is also important to note that PNA-FISH, unlike all single-cell sequencing and qPCR techniques, provides a copy number assessment without the use of genomic amplification.

We have also now addressed the regions in the text that required clarification for our use of PNA-FISH.

*In summary, the conceptual basis of these studies is of interest. However, by repeatedly reciting that they have proved the functional significance of their observations by five different methods instead of candidly addressing the limitations of their studies and the possibility that their data may have no bearing whatever on the mechanistic basis of sporadic Alzheimer's disease, I find the manuscript unsatisfactory for publication in its present form*.

In response to the reviewer comments we have toned-down our assertion that *APP* copy number increases are a pathogenic mechanism and added a critical discussion assessing the limitations of our study. We, like the reviewer, are very interested in the mechanism of mosaic changes, however this is well beyond the scope of this study. Our study is primarily descriptive in nature, as are virtually all studies limited to working with post-mortem AD brains that reflect a progeric human disease requiring decades to manifest. Importantly, any novel process or gene that may underlie mosaicism, while enormously interesting, could not be proven to underlie sporadic AD without decades of prospective human population and/or clinical trials to test a relevant hypothesis. Therefore, requests for a molecular mechanism beyond validated *APP* represent, in our opinion, an unreasonably high bar for publication on a new phenomenon as we report here. As noted in the second major point identified in the cover letter from the editors, we have toned-down our claims by highlighting the possibility that alterations in DNA content variation and increased *APP* copy number may not be a cause of sporadic AD.

Reviewer #2:

*[…] Through their work the authors provide convincing evidence that increase in DNA content, which appears to be due to genomic amplifications in many loci including the* APP *gene locus, is a common phenomenon in the cortex, less common in the cerebellum and uncommon in lymphocytes. This is very interesting and indeed provides a possible mechanism for the pathology of sporadic AD. However, no significant evidence is shown to support that this phenomenon is more prevalent in the AD cortex than controls. The presented data do show a somewhat more pronounced effect in AD but the similar, albeit weaker, finding in controls is downplayed, and the data is mostly placed in the supplementary material. Statistical comparisons between case and control brains that could convince the reader that this is indeed more pronounced in AD brains are not provided and would likely not be significant due to the sample size.*

We thank the reviewer for these suggestions and, as discussed at the start of this rebuttal, have now added new experimental data consisting of new brain samples as well as re-analyses of previously published data (meta data) that, taken independently or combined together, increase the statistical significance values (P=0.0001) between non-diseased and AD samples; these were noted above in response to the first major issue of the cover letter, and new data are provided in the new Figures 2 and 3 (old Figure 2 and Figure 2—figure supplement 1). Our new data demonstrate statistically significant differences between case and control.

*While it is likely that there is a true difference, the authors should be more cautious about making statements of “mosaicism in AD brains”, which implies that this is not the case in controls. They should also provide more discussion on the fact that this is also observed in cases. While this phenomenon provides a substrate that might be necessary or just facilitate the development of plaques in AD, based on the authors own data it is likely not sufficient. These points should be made in the Discussion*.

We thank the reviewer for this critical assessment of the data and we have toned down our statements in our revised Discussion, as noted in response to the other reviews.

*The manuscript would also benefit from more discussion on the significant differences between lymphocytes, cerebellum and cortex. It seems perhaps counter intuitive that cells that undergo many divisions throughout life maintain their DNA content much better than cells that do not, and some discussion on this would be useful in better understanding the results and their potential impact*.

We have now included further discussion on this topic. DNA content changes have been examined in a limited number of tissue types, cell types, and conditions. It is important to note that the interrogated neurons are likely terminal in their fate: they will not divide further, contrasting with lymphocytes. Populations of cells, like lymphocytes “that undergo many divisions” are those that have been selected to be capable of continuing cell division, as seen in stem and progenitor populations, which are not found amongst neurons that are, by definition, post-mitotic. We think it is likely that many dividing cells that deviate significantly from the germline genome—and thus would be incapable of further division—are eliminated by cell death and would not be in the population interrogated for DNA content. Support for this view includes our reports on aneuploidies in neuroprogenitor cells (51; 6, 7), whereby the most extreme forms appear to be eliminated by caspase-mediated cell death.

Reviewer #3:

*[…] While the authors provide a set of compelling evidence using state of the art approaches to support their conclusion, one assumption is that the increase of up to 12 copies of* APP *gene in neurons from AD are all functional and active genes. As previous work has documented that the coding region of human APP gene spans ∼400 kb of genomic DNA (Rooke et al., Mamm. Genome, 4:662-9, 1993), it is possible that the increase in* APP *gene copy number in neurons from AD cases do not represent active full length* APP *genes. Direct evidence to distinguish whether active full length copies of* APP *gene from inactive non-transcribed pseudogenes account for the increase in gene copy number in neurons will strengthen this manuscript. For example, is there any evidence that the* APP *transcript (as judged by* in situ *rtPCR using RNAscope technology) is increased in these neurons with elevation in copy number of* APP *gene? Positive data will strongly support the notion that the increase in amyloid burden can be attributable to the increase in number of active* APP *gene in neurons within affected regions of sporadic AD patients.*

We absolutely agree that these data would be of high interest, but also note several technical and conceptual issues that we believe limit the feasibility and interpretation of the experiment.

1) Genetic linkage of *APP* gains in some forms of familial AD and Down syndrome are clear, however the mechanisms through which this gain produces AD remain unclear, including at a transcriptional level. Our data in fact argue that there are likely to be myriad additional changes to the genome and genes produced by cortical increases in DNA content that represent, on average, hundreds of megabases of additional sequence. It is also a fact that all forms of AD are progeric: even in familial cases, it takes decades to manifest despite the germline presence (for familial disease) of the “causative” gene mutation. We believe that it is likely that *APP* gene expression will be increased at some point in the life cycle of an affected AD neuron, however this could occur early in life, or perhaps at the terminal phase of neuronal life—AD is a neurodegenerative disease—which might prevent us from interrogating those neurons with high levels of gene transcription since they would be eliminated by death. This further raises the issue of when, as well as where, one should look for transcriptional changes. It is further possible that there are heterogeneous responses of individual neurons to increased *APP* copy number that may or may not involve increased gene transcription. Moreover, we also see increases in *APP* gene copy number in rarer non-diseased neurons, raising the issue of how to interpret the obtained data from AD neurons, especially in a statistically significant manner related to pathogenesis. At this juncture, the genetic linkage with familial AD combined with our mosaic data provides a plausible explanation, with both disease forms awaiting cell and molecular mechanistic detail.

2) It is also notable that sporadic AD has an onset decades later than the changes seen in Down syndrome, which contains a single additional copy (in all cells). We are unable to find reports of quantitative *APP* RNA expression in single Down syndrome cells, but are certain there is not uniform expression in all cells that contain the additional copy, underscoring issues of how one interprets the presence or absence of a transcriptional signal in neurons with varied *APP* copy number.

3) As importantly from a technical standpoint, it is not currently possible to identify simultaneously, in a single AD neuron (which is the appropriate goal), genomic amplification of *APP* and mRNA from the same locus. Technological approaches in the future should allow this to be realized.

4) RNAscope is an intriguing idea, however recent discussions with the manufacturer indicate that its application for genomic loci is unproven, as is its use for RNA within a nucleus that contains perhaps 5% or less of the mRNA that one would want to interrogate. That said, future studies may well provide new insights via RNAscope. In our hands, in situ PCR has not proven reliable or quantitative, and this is not a technology that we currently view as appropriate for the questions at hand.

5) Additionally, the full length *APP* gene is ∼298 Kb, while the two exons used in this study are separated by ∼170 Kb. Using single cell qPCR on the Biomark HD, a concordance rate of ∼85% was seen between the two exons. This suggests that, minimally, most of the gene is amplified. How this relates to individual neurons at distinct points of their life cycle and location within the brain (as well as specific neuronal subtype) again remains for future work.

Thus, we are enthusiastic about future analyses stemming from the observations of this manuscript, but believe that garnered information as proposed in this critique would not change the basic result, and obtaining such data would require a very significant effort beyond the scope of the current study.